# Mapping Land Use Land Cover Transitions at Different Spatiotemporal Scales in West Africa

**Beatrice Asenso Barnieh [1,2]**, **Li Jia [1,\*]**, **Massimo Menenti [1,3]**, **Jie Zhou [4]** and **Yelong Zeng [1,2]**

1   State Key Laboratory of Remote Sensing Science, Aerospace Information Research Institute, Chinese Academy of Sciences, Beijing 100101, China; b.a.barnieh@radi.ac.cn (B.A.B.); m.menenti@radi.ac.cn (M.M.); zengyl2018@radi.ac.cn (Y.Z.)
2   University of Chinese Academy of Sciences, Olympic Campus, Beijing, 100101, China
3   Delft University of Technology, Faculty of Civil Engineering and Geosciences, Stevin Weg 1, 2825 CN Delft, The Netherlands
4   Key Laboratory for Geographical Process Analysis & Simulation of Hubei Province, College of Urban and Environmental Sciences, Central China Normal University, Wuhan 430079, China; zhou.j@mail.ccnu.edu.cn
\*   Correspondence: jiali@aircas.ac.cn

**Abstract:** Post-classification change detection was applied to examine the nature of Land Use Land Cover (LULC) transitions in West Africa in three time intervals (1975–2000, 2000–2013, and 1975–2013). Detailed analyses at hotspots coupled with comparison of LULC transitions in the humid and arid regions were undertaken. Climate and anthropic drivers of environmental change were disentangled by the LULC transitions analyses. The results indicated that human-managed LULC types have replaced the natural LULC types. The total vegetation cover declined by −1.6%. Massive net gains in croplands (107.8%) and settlements (140%) at the expense of natural vegetation were detected in the entire period (1975–2013). Settlements expanded in parallel with cropland, which suggests the effort to increase food production to support the increasing population. Expansion of artificial water bodies were detected in the humid regions during the period of 1975–2000. Nonetheless, shrinking of water bodies due to encroachment by wetlands and other vegetation was observed in the arid regions, coupled with net loss in the whole of West Africa. The results indicate deforestation and degradation of natural vegetation and water resources in West Africa. Underlying anthropic drivers and a combination of anthropic and climate drivers were detected. LULC transitions in West Africa are location specific and have both positive and negative implications on the environment. The transitions indicate how processes at the local level, driven by human activities, lead to changes at the continental level and may contribute to global environmental change.

**Keywords:** West Africa; arid region; humid region; LULC transitions; climate; anthropic

## 1. Introduction

### 1.1. Environmental Change in Africa

Africa, especially the Sahel region, is often described in scientific literature as the hotspot of global environmental change. This stems from the demographic evolution, intensive anthropic/human activities, and the severe recurrent drought spells observed in the 1970s and 1980s [1–3]. The aforementioned environmental disturbances metamorphosed into land degradation and threatened vegetation growth and food security in this continent [4]. These are environmental issues of global concern but they are being questioned in the literature based on documented cases of vegetation recovery (re-greening) after the aforementioned drought [5–10]. Addressing these issues requires information about the nature, the extent, and the causes (driving factors) of the environmental changes

in Africa [9–12]. Several attempts have been made by a growing body of scientists to unravel the aforementioned information [5–10].

Vegetation trend analyzes with normalized difference vegetation index (NDVI) and other vegetation indices are the frequently used methods for the assessments of the above-mentioned environmental changes in Africa. Some evidence from such analyses points to re-greening (vegetation recovery) of Africa, predominantly in the Sahel region, after the severe drought [5–10]. Opposing evidence by other research scientists has also been documented [9–12]. The arguments are that, even if there had been instances of re-greening, there are some areas where browning (vegetation loss) had been reported [9–12]. The debate is still on-going and mostly likely, due to the over reliance on trend analyses of vegetation conditions captured by vegetation indices, e.g., NDVI, for vegetation trend assessments. The complete information about the nature, extent, magnitude, and the major driving factors of the environmental change cannot be fully extracted from the vegetation trend analyses alone. Environmental change assessments with trend analyses of vegetation conditions do not take full account of the underlying processes, such as how vegetation cover may be replaced by built-up areas, e.g., human settlements and vice versa over a period of time [12,13].

According to Rasmussen et al. [12] and Brandt et al. [14], it is unclear whether the observed re-greening or browning of vegetation cover is attributed to an increase or decrease in woody vegetation, herbaceous vegetation, or crop cover. Furthermore, Rasmussen et al. [12] underscored that little quantitative information is available on the state, rate, and drivers of change in woody vegetation cover at the continental scale of Africa. The vast majority of findings based on vegetation index assessments, for example the findings by Hickler et al., Seaquist et al., and Huber et al. [15–17], further suggest that climate and natural variations such as variations in annual rainfall patterns and soil moisture are the major driving factors of the observed vegetation dynamics and the subsequent environmental change in Africa. In contrast, current evidence at the continental scale of Africa suggests that the aforementioned changes cannot be fully explained by climate factors alone [5,9,10,13,14,18–20].

Numerous studies at the national and subnational levels also point to a combination of climate factors, such as variation in annual rainfall pattern, and anthropic driving factors, such as human activities in the form of cropland expansion, settlement expansion (urbanization), development of artificial water bodies, firewood extractions, timber logging, and many others as the possible driving factors of the environmental changes in Africa [21–27]. Several field surveys at some local level in Africa also attribute the observed environmental changes such as deforestation to anthropic factors or human activities [25,28–35]. Brandt et al. [14] linked the intensive human activities and the quest to expand agricultural fields to ensure food security at the expense of natural vegetation, biodiversity, and carbon stock enhancement with the recent population growth in Africa.

Disentangling the climate driving factors from anthropic factors is vital for natural resource managers and planners to ensure efficient allocation of resources and spatial targeting of climate change adaptation and mitigation programs such as the "Reducing Emissions from Deforestation and Forest Degradation (REDD+) Project" [36–38] and the "Great Green Wall Initiative" [39–41]. This is crucial in Africa, especially the Sahel region, where a review of historical rainfall patterns indicated that future drought is likely to occur in less than two decades [42].

Nonetheless, Helldén [43] noticed that there is a synergy between climate and anthropic drivers of environmental change in Africa. Assessing the effect of each driver requires supplementing the vegetation trend analyses with a series of LULC change analyses over a long period of time [44]. Mbow et al. [44] gave an overview of the indicators that may be employed in earth observation (EO) and geo-information science (GISc) to assess land degradation and their drivers. Among the indicators, the LULC change indicator has the advantage of separating the different LULC types, e.g., forestland, herbaceous vegetation, shrubland, cropland, and bare land settlements, into distinctive categories and may serve as a proxy for identifying the drivers of environmental change in Africa.

### 1.2. LULC Transitions and Underlying Driving Factors

Lambin et al. [45] defined LULC transitions as socio-economic and biophysical changes associated with environmental change. According to Lambin et al. [45], these changes may originate from a set of interconnected changes that reinforce each other but take place in many different components of the earth system (e.g., biosphere, hydrosphere, anthroposphere). Lambin et al. [45] further emphasized that LULC transitions in each region depend on the possible developmental pathways where the direction, size, and speed of changes can be influenced by socio-economic and environmental factors such as policy specific circumstances, population growth, poverty rate, changes in human lifestyle, and biophysical parameters.

Understanding the long-term LULC changes/transitions and the underlying driving factors in Africa may serve as a springboard to identify and separate the effect of climate drivers (e.g., variation in rainfall pattern) of environmental change from anthropic drivers, e.g., expansion of human settlements and croplands [9,10,19]. This is critical in Africa, where it has been established at the local level that anthropic factors may play a central role in land degradation/environmental change, and it is a big challenge to disentangle the effects of anthropic drivers of environmental change from climate drivers [43].

Verburg et al. [46] highlighted that LULC changes/transitions analyses may offer useful information about the trends in land degradation, desertification, biodiversity loss, and deforestation in a given region, as well as the interplay of socio-economic and ecological drivers. LULC changes/transitions analyses may also deepen our understanding of the transitions occurring on different landscapes over a period, such as how forest cover, other vegetation cover, water bodies, and wetland may transition into cropland and built-up areas. Assessment of historical trends in LULC changes/transitions at multiple time intervals and the quantification of its impact on the ecosystem are also required for planning multiple uses for the natural resources on the land.

Monitoring and assessing LULC changes/transitions rapidly on a global and continental scale can be done effectively by applying geo-information system (GIS) technologies such as change detection [47] algorithms on remotely sensed (RS)/EO datasets. RS datasets are widely accepted for change detection due to their spatial continuity, high temporal frequencies, and wide choice of spatial and spectral resolutions. The rationale for change detection in RS includes identification of spatial location of the surface feature under investigation, examining the nature of changes, quantifying the magnitude of changes, and so forth [48–50]. Nevertheless, Karlson and Ostwald [51] stressed that LULC changes and the quantification of their impacts on the ecosystem have been poorly mapped and have not received much attention at the continental scale of Africa, though the approach has been more successful at the local scale of Africa [23,25,29–33,35,52–57].

One of the few insightful LULC change analyses at the continental scale of Africa is the sample-based LULC change analyses by Vittek et al. [58]. This study analyzed Landsat MSS/TM satellite imageries from 1975–1990. The sample-based LULC change monitoring over 25 years in Sub-Saharan Africa presented by Brink and Eva [59] also provided useful information about the LULC change, but considered only the period of 1975–2000. This may be due to the difficulty in acquiring time series of continental EO/RS and GIS LULC datasets. The aforementioned LULC transitions analyses by Brink and Eva [59] and Vittek et al. [58] served as a starting point to monitor long term environmental change in Africa after the severe drought of the 1970s and to assess the impact of human activities on the environment, but the LULC classification scheme they adopted limited the opportunity to link changes with climate/natural and anthropic drivers. For example, human-induced LULC changes, such as cropland and settlement expansions, may serve as indicators of anthropic drivers of environmental change. However, these two LULC types were not defined as separate classes in their LULC transitions analyses at the continental and subcontinental scales of Africa and West Africa, respectively, making it difficult to link changes with underlying climate and anthropic drivers.

The temporal coverage of the LULC change analyses by Brink and Eva [59] and Vittek et al. [58] were 1975–2000 and 1975–1990, respectively. This limited the opportunity to understand the LULC transitions

after the period of 1990s and 2000s in the latter and the former, respectively. Moreover, the studies by Brink and Eva [59] and Vittek et al. [58] did not provide information about the states of the wetlands and water bodies. Such information is vital for assessing the impact of LULC change on water balance, biodiversity, soil fertility, and land degradation [47]. "Wall-to-wall" mapping of LULC change and transition analysis is a reliable way to capture the impact of LULC change on the natural environment over a longer period of time [48–50]. The previous LULC change analyses in Africa by Brink and Eva [59] and Vittek et al. [58] applied a sample-based approach for monitoring the LULC changes and cited the large amount of satellite data required for developing global and continental LULC datasets as well as the long processing time as the major limitations of historical LULC transitions mapping in Africa. The sample-based approach for monitoring LULC changes may miss some location specific LULC change information.

Against this background, "wall-to-wall" mapping of historical LULC transitions with long-term series global and continental LULC datasets, comprehensive analyses to understand the nature of these LULC transitions after the severe drought of the 1970s and 1980s, as well as the relative impact of climate and anthropic drivers on the environment are urgently needed at different spatiotemporal scales in Africa to ensure sustainable management of natural resources. Kganyago and Mhangara [60] and Mhangara et al. [61] highlighted the important roles EO data and GIS applications may play in achieving the United Nation's (UN) 2030 sustainable development goals as well as the Africa Union's sustainable environmental goals, embedded in the Agenda 2063 strategic development targets in Africa. According to Kganyago and Mhangara [60] and Mhangara et al. [61], some African countries such as Nigeria, Egypt, Kenya, and South Africa have already made some major strides in the adoption of EO and geospatial technologies for ensuring successful implementation of the sustainable development goals.

### 1.3. Global and Continental EO LULC Datasets

Current innovations in RS and GIS application have laid the foundation for LULC change analysis at the continental scale of Africa and at the global level. The emergence of advanced RS and GIS datasets' acquisition and accessibility, high performance computing capability, in combination with high leveled image classification and change detection techniques, as well as crowdsourcing platforms such as Google Earth Engine (GEE) and Geo-Wiki, permit easier and quicker data integration and development of LULC datasets over large area and even at the global scale. Therefore, LULC change detection and transitions analyses at the continental scale have been made easier [47–50,62–68].

The current "wall-to-wall" LULC data mapped at 2 km spatial resolution at the subcontinental scale of West Africa at three time intervals (1975, 2000 and 2013) released by the United States Geographical Survey (USGS), the West African Land Use Dynamic project [69], is one of the major advances in LULC change analysis. These LULC data delivered an insightful history of LULC change at this spatial scale [69]. Other valuable open-source continental and global LULC datasets that are useful for large scale LULC transitions and driving factors assessments are:

- The ESA CCI-LC 300 m LULC datasets [70] produced by the European Space Agency (ESA) Climate Change Initiative (CCI) project. These LULC datasets were mapped at an annual interval over 24 years, spanning from 1992 to 2015. More information about how to download the datasets can be found in the user manual http://maps.elie.ucl.ac.be/CCI/viewer/download/ESACCI-LC-QuickUserGuide-LC-Maps_v2-0-7.pdf of the datasets.
- The MODIS LULC datasets [71], produced from moderate resolution imaging and spectro-radiometry and mapped at 500 m spatial resolution at one year intervals. The (MCD12Q1) version is available for downloads at https://doi.org/10.5067/MODIS/MCD12Q1.006.
- The Globe Land Cover (GLC-30m) LULC datasets produced by the National Geomatics Center of China [72] and regarded as the first global 30 m resolution dataset for two time periods (2000 and 2010). The datasets can be accessed at http://www.globallandcover.com/GLC30Download/index.aspx.

In this study, historical LULC transitions in the periods of 1970–2000, 2000–2013, and 1975–2013 as well as the relative impact of climate and anthropic activities on natural environment in West Africa and two sub-regions (humid and arid) were evaluated. The overarching objectives were:

- To examine the extent, magnitude, and nature of LULC transitions during 1970–2000, 2000–2013, and 1975–2013 in West Africa and two sub-regions (humid and arid) with long-term series open-source LULC datasets.
- To disentangle natural/climate and anthropic drivers of environmental change in West Africa with LULC transitions mapping and underlying driving factors analyses.

*1.4. The Study Approach*

We hypothesized that anthropic drivers had significant impact on the LULC dynamics in West Africa after the severe drought of the 1970s and 1980s. The hypothesis was tested by extracting useful earth surface spatial information from EO/RS open-source big data spanning a long period with GIS applications. As mentioned before, many up-to-date open-source LULC data are currently available at the continental and global scales. However, satellite imagery from different sources and different approaches were used to generate these LULC data. According to Tsendbazar et al. [66], LULC statistics from different sources are inconsistent. One of the focuses of this study was to examine the LULC transitions during the drought period in West Africa and compare it with the transitions after the drought. We employed the long time series USGS LULC datasets for this analysis, because, despite the fact that these datasets are "wall-to-wall" [69], currently, it is the only EO LULC datasets that cover the temporal resolution of the drought period (i.e., 1970s) in West Africa, and hence they are more suitable to detect historical LULC transitions during the severe drought period.

Furthermore, forestland and cropland estimates from the USGS LULC map in the year 2000 had been evaluated against Food and Agriculture Organization (FAO)-Statistics (FAOSTATS) [73] and other global LULC maps by our research team. We found that the forestland and cropland estimates of the USGS LULC data [69] agreed better with the FAOSTATS as compared to the other global LULC data (Unpublished results by the authors of this study). Additionally, the USGS datasets at the three time periods [69] captured 24 LULC classes/types (including wetlands and water bodies) that were not assessed in the previous LULC transitions analyses in Africa by Brink and Eva [59] and Vittek et al. [58] but are vital in understanding the impacts of LULC change and human influence on water balance and wetlands restoration in West Africa.

To examine the impact of anthropic drivers on the LULC dynamics, we aggregated and reclassified the 24 LULC classes in the "wall-to- wall" USGS LULC data generated by CILSS [69] into seven distinctive LULC classes. Human-managed LULC types and natural LULC types were very clear after this process. A post-classification change detection algorithm [47] was applied to the original and the reclassified LULC data to trace the nature, extent, and magnitude of the LULC transitions in West Africa, two sub-regions (humid and arid) and some local hotspots in three time periods (1975–2000, 2000–2013, and 1975–2013). To examine the underlying drivers of the LULC transitions and the anthropic impacts, findings from the transitions analyses were supplemented with a literature review of the study area to disentangle climate drivers from anthropic drivers.

In this study, we demonstrated that climate and anthropic impacts on LULC transitions and environmental change in West Africa during the periods of 1975–2000, 2000–2013, and 1975–2013 varied through time and space. The findings from this study will broaden the current understanding of human interaction with the natural environment and its implications on the ecosystem as well as the underlying driving factors of environmental change after the severe drought of the 1970–1980s in West Africa and the two sub-regions (humid and arid).

This paper is organized into six sections. After this first section (introduction) is the second section, which describes the study area and the data used. The methodology developed for this research is presented in Section 3. The results are shown in Section 4. The discussions are presented in Section 5, and they are based on the four major issues (the environmental change in Africa, LULC transitions and

underlying driving factors, global and continental EO LULC datasets, and the study approach) raised in this section. The paper ends with some major conclusions in Section 6.

## 2. Study Area and Datasets

### 2.1. Study Area: West Africa

The study area covers Sub-Saharan West Africa between 4° N–18° W and 18° N–24° E (see Figure 1). West Africa's landscape can be described as a complex ecosystem that encompasses a wide diversity of landscapes, ranging from alluvial valleys in Senegal and Ghana, sandy plains and low plateaus across the Sahel and undulating hills in Togo, to rocky mountains reaching over 1500 m in Guinea and 1800 m in Niger. The extent of this subcontinent is about $8 \times 10^6$ km$^2$ and constitutes about a quarter of the entire size of Africa. The subcontinent is characterized by its distinctive natural features (geology, relief, climate, vegetation, and soil) and these have shaped the lifestyle and the land use of the people living in this region. It is common to observe pastoral activities in the far north and crop farming in the southern part. The area includes ancient Pre-Cambrian rocks, which over the years have developed into massifs and highlands such as the Tibesti mountains, Adrar des Ifohas, and Fouta Djallon. In general, the relief of West Africa is relatively flat and low [74].

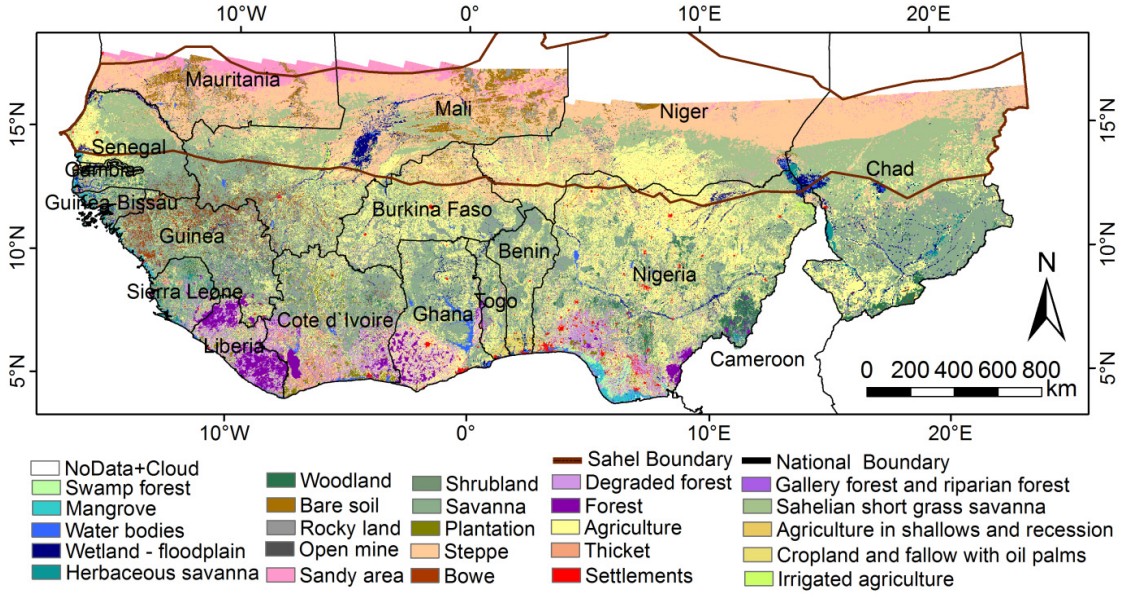

**Figure 1.** Map of the study area.

West Africa is also noted for its unique water bodies such as the Niger River, well-known as West Africa's longest river, which originates in the Guinea highlands where rainfall is high; the Senegal River; Lake Chad Basin; and Volta Lake in Ghana, which is regarded as the world's largest artificial lake. The frequency of rainfall in most of the countries in West Africa (Southern Sahara to the humid coast) is restricted to only one season, which may last from one to six months. The southern portions of the coastal countries are characterized by two rainy seasons, of which one is well-known for its longer duration and the other by a shorter duration [75].

The study area is characterized by five broad bioclimatic zones (Saharan, Sahelian, Sudanian, Guinean, and Guineo-Congolian), which are defined by a distinctive variation in the rainfall pattern and the vegetation varying along the rainfall gradient from 0 to 150 mm in the Sahara region to as high as 2200–5000 mm in the Guineo-Congolian region [74]. According to Church [74], the maximum temperature range also increases with latitude in West Africa. In the humid south, the temperature has a limited yearly variation, while in the arid north, the daily temperature may range from 0 °C to more



than 45 °C. The vegetation in the Sahara region is scanty, except in depressions and oases where water is available.

In the Sahel region of West Africa, the vegetation is generally characterized by open herbaceous types, i.e., steppe and short grass savannah often mixed with woody plants. The Sahel region is also noted for the thorny trees of the genus acacia and annual grasses. The northern region is dominated by sparse vegetation cover, ranging from open grass and shrubland mainly used for grazing, whilst the southern region is characterized by a larger amount of evergreen and semi-deciduous dense vegetation cover, woodland, and savannah [76]. Large plains, sand dunes, and rocks are common occurrences in the Sahel region of West Africa. Small water ponds for irrigated agriculture are now widespread in this region. Subsistence farming, alongside livestock rearing, is the dominant land use in the Sahel region of West Africa. Cereals such as sorghum and millet are the major crops in the Sahel [77]. Mauritania, Niger, northern parts of Senegal, Mali, Burkina Faso, Chad, and Nigeria constitute this biome in West Africa [4]. The extent of this biome in the study area is demarcated with a brown outline in Figure 1. The remaining portion of the map constitutes the Sudanian, Guinean, and Guineo-Congolian regions of the study area from north to south, respectively (see Figure 1).

The Sudanian region of West Africa is dominated by vegetation, which ranges from open tree savannah to wooded savannah and open woodland. The grassland here is often taller than in the Sahel region. The vegetation in the Guinean region is dominated by wet-and-dry deciduous or semi-deciduous forest with dense and closed forest canopy, which often forms an understory with very high trees. The vegetation in the Guineo-Congolian region is rich and dense forest with trees reaching over 60 m with intermingling crowns. The biodiversity in this region is considered the richest in West Africa [69].

*2.2. The Datasets: USGS LULC Maps in 1975, 2000, and 2013*

The LULC datasets we used for the analyses were produced by the United States Geographical Survey (USGS) West African Land Use Dynamic project [69] and consist of 24 LULC types mapped in 1975, 2000, and 2013. These data cover seventeen countries in West Africa at 2 km spatial resolution, except Chad, which was mapped at 4 km, Gambia and Cape Verde, which were mapped at 1 km and 500 m, respectively (see Figure A1 in the Appendix A). The final spatial resolution of the datasets available for download was 2 km. We did not consider the Cape Verde islands, which were not mapped in 1975, in our analyses. Moreover, according to CILSS [69], the northern parts of Mauritania, Mali, Niger and Chad were also not mapped, because these areas are within the Sahara desert and exhibit stable vegetation and other LULC types (e.g., sand and rocks) through time. Therefore, the LULC changes and transitions analyses presented in this paper exclude the unmapped northern parts of the aforementioned countries and Cape Verde. The white regions on the map in Figure 1 are the areas without data and therefore were not mapped.

According to Tappan et al. [78], the three LULC maps with 24 land cover classes are based on the Yangambi classification system as well as other commonly used classification systems in West Africa. Images from the Advanced Space Borne Thermal Emission and Reflection Radiometer (ASTER), Landsat TM, together with field data, Google Earth images, thousands of aerial photographs, and very high resolution images were used to produce these datasets by the help of a visual photo interpretation tool, i.e., Rapid Land Cover Mapper Software. Tappan et al. [78] clarified that the LULC datasets were validated by thousands of the aforementioned aerial photographs and high resolution satellite images. Additionally, independent reviews of the datasets were further undertaken by the USGS and the respective country team of image interpreters. The comprehensive definitions of the various LULC types in the datasets can be referred from CILSS [69]. The datasets are available at https://eros.usgs.gov/westafrica.

## 3. Methodology

### 3.1. The General Approach

The LULC transitions analyses were undertaken by first detecting the general LULC changes in terms of relative net gains and losses in the area covered by the 24 LULC classes/types in the original USGS LULC datasets [69] from 1975 to 2013 with post-classification change detection technique [47]. This was useful for understanding the LULC transitions and to trace changes in some specific LULC types. For example, changes in specific LULC types such as expansion of settlements, irrigated cropland, open-mines fields, artificial water bodies and plantations in the original 24 LULC classes were linked with anthropic drivers.

The 24 LULC classes in the original LULC data [69] were then reclassified into 7 LULC classes, i.e., cropland, forestland, other vegetation, wetland, water bodies, settlements and other LULC types, by aggregating similar classes in the data into broader classes. Details of the aggregation and re-classification can be found in Table 1 in Section 3.2. The aggregation and the new classification scheme we developed segmented the maps and allowed a clearer distinction between the natural LULC types (e.g., forestland and other vegetation) and the human-induced LULC types (e.g., cropland and settlement). This was useful for disentangling climate/natural drivers from anthropic drivers of the LULC changes/transitions in the study area. Such a distinction was difficult to identify in the previous LULC changes/transitions analyses at the continental scale by Brink and Eva [59] and Vittek et al. [58]. Additionally, the detailed transitions analyses would have been daunting and unclear with the 24 LULC classes.

**Table 1.** The Land Use Land Cover (LULC) reclassification scheme developed for the USGS datasets in 1975, 2000, and 2013 at 2 km spatial resolution.

| USGS Original LULC Types | Reclassified LULC Types Developed by This Research |
|---|---|
| Rain-fed agricultural land, plantation, agricultural land in recession, irrigated agricultural land, cropland in shallows with oil palm. | Cropland |
| Forest, gallery forest, degraded forest, swamp forest, woodland, mangrove. | Forestland |
| Savannah, steppe, bowe, thicket, herbaceous, Sahelian short grasses. | Other vegetation |
| Wetland | Wetland |
| Water | Water |
| Settlement | Settlement |
| Rocky land, sandy areas, bar soil, open mines | Other LULC |
| Cloud cover, no data | No data-cloud cover |

The previous LULC changes/transitions analyses in Africa by Brink and Eva [59] and Vittek et al. [58] used a sample-based approach at only one time interval, (1975–2000) and (1975–1990), respectively, and did not include water bodies and wetlands assessments in their analyses. We filled this gap by including assessments of water bodies and wetlands in our analyses. This was central to understanding the impact of availability of water bodies and water balance in the LULC transitions and to identify new developments in the form of dams and small reservoirs, which are indicators of anthropic activities.

We detected the extent, magnitude, and the nature of the LULC transitions of the newly reclassified LULC types with post-classification change detection technique [47] at three time intervals (1975–2000, 2000–2013, and 1975–2013). This allowed us to trace the relative impacts of climate and anthropic drivers of environmental change in different periods, thereby capturing the LULC response to changes

in the drivers over time. Statistics on relative net gains and losses of the seven aggregated LULC classes were obtained by estimating the absolute and relative changes in the area of each LULC class in 1975–2000, 2000–2013, and 1975–2013. This answered the research questions about the nature, extent, and magnitude of LULC change after the severe drought of the 1970s and 1980s in West Africa. Such information could not be retrieved from the previous vegetation trend analysis in West Africa (e.g., [12]). The LULC transitions in the arid and the humid regions of the study area were compared at three time intervals. This was also useful to capture and compare differences in the LULC transitions, which may be caused by location specific socio-economic and ecological factors [45].

We identified hotpots of the major LULC transitions by applying a majority filter to a moving 5-by-5-pixel window [79–81] on the LULC change map, we obtained from the LULC change analyses. This served as an objective and novel approach for identifying the major hotspots of LULC transitions and drivers of environmental change in West Africa. Such information is vital for policy interventions related to the environment, such as the "REDD+ Policy" [36–38] and the "Great Green Wall Initiative" [39–41], which require targeting hotspots with severe deforestation rates or environmental change in Africa.

The possible underlying processes of the observed LULC transitions in some hotspots of the study area were documented by a literature review. The findings from the hotspot analysis and the literature review allowed us to link the major drivers of the LULC transitions with climate and anthropic drivers of environmental change in West Africa. In this context, climate and anthropic drivers of environmental change in the study area were disentangled by human-induced LULC types such as expansions of settlements, rain-fed and irrigated croplands, open-mine fields, new developments of artificial dams, and reservoirs we detected from the LULC transitions analyses.

### 3.2. Reclassification and Post-Classification Change Detection

The purpose of LULC changes and transitions analyses differs from different study domains. As a result, LULC classification schemes/legends are developed based on the intended application of the data. In view of this, several legends (e.g., FAO Land Cover Classification system (LCCS), International Geosphere–Biosphere Programme (IGBP), BIOME Biogeochemical Cycles (BGC), University of Maryland legend (UMD), Leaf Area Index (LAI), and so forth) of global LULC datasets are inconsistent. This limits comparison of LULC estimates from different global and continental LULC datasets [66,71]. At the data development stage, the FAO Land Cover Classification system (LCCS) is recommended as the standard. However, after the development, the various LULC datasets may be modified to serve the intended purpose or application. The legend of the original USGS LULC maps (1975, 2000, and 2013) with 24 LULC types/classes [69] we used for this study is a hybrid between Yangambi classification system and other commonly used classification systems in Africa, such as the FAO Land Cover Classification system (LCCS) [78]. Therefore, to allow the LULC estimates from this study to be comparable with the existing open-source global LULC datasets, we aggregated the 24 LULC classes in the original USGS LULC maps [69] into seven broader classes, i.e., cropland, forestland, other vegetation, wetland, water, settlement, and other LULC types, and reclassified them.

The reclassification was undertaken by grouping similar small classes such as the sub-classes of forestland (e.g., forest, gallery forest, degraded forest, swamp forest, woodland, and mangrove) and agricultural land (e.g., rain-fed agricultural land, plantation, agricultural land in recession, irrigated agricultural land, and cropland in shallows with oil palm) in the original dataset with 24 LULC classes into one broader forestland and cropland class, respectively. All the other vegetation types apart from the classes defined as forestland and cropland in our study were also grouped into one broader class: "other vegetation". "Settlements", "wetlands", and "water bodies" in the original dataset were retained as separate classes in the new reclassified data. All the other remaining classes were further grouped into one broader class: "other LULC types". The same legend was used to reclassify some existing global and continental LULC data (i.e., GLC-30m, MODIS-MCD12Q1, and ESA-CCI-LC) in a previous study by the authors of this study (unpublished results). This allowed a comparison of the

LULC estimates from the existing data with the USGS data. Table 1 shows the reclassification scheme developed by this research and how the 24 LULC classes were assigned into new classes.

The 24 LULC classes in the original USGS maps were first analyzed to detect the general trend, i.e., the direction of LULC change (positive or negative) in the individual classes from 1975 to 2013 by performing post-classification change analysis [47] with the "combine tool" in ARCGIS Software Program (version 10.3.1). The "combine tool" combines multiple raster datasets and assigns a unique output value to each unique combination of input values [80]. The output is a new raster map with an attribute table that gives information on the changed and unchanged LULC classes as well as the transitions that occurred over the periods of the analyses.

The relative net gains/losses (%) in each LULC class were calculated by first applying a change detection algorithm to the two LULC maps (1975 and 2013) to obtain gross losses and gains in area extent of each LULC class [48–50,58]. The results were processed and the outputs were divided by the area extent in 1975 (i.e., the initial LULC area extent). The results were multiplied by one hundred to get the relative net changes (losses and gains) in percentage. It must be noted that NoData–Cloud Cover in the original USGS maps (see Figure A1 in the Appendix A) with 24 LULC classes was not corrected by CILSS [69]. In the seven aggregated LULC classes we developed for the detailed transitions analyses, which will be described later, the effect of NoData–Cloud Cover on the transitions analyses was adjusted by masking their pixels from the analyses [82]. Additionally, the 2013 LULC map, which covered Cape Verde, contained shrubland as one of the LULC classes, i.e., 25 classes were mapped in 2013, but after excluding Cape Verde from the analyses, the 2013 LULC map reverted to 24 classes just like the other periods.

The same procedure [47] we employed for analyzing the LULC transitions for the 24 LULC classes from 1975–2013 was used to analyze the LULC transitions in the three reclassified LULC maps with seven classes, representing the three different time periods (1975, 2000, and 2013). Here, three different combinations of LULC maps were generated, namely, the combination of the LULC maps of the year 1975 with 2000, 2000 with 2013, and 1975 with 2013. These three combinations represent the three periods of our LULC change analyses. A change in a given LULC class in each period was characterized by a "loss" or a "gain" as follows: Maps of the loss and gain areas for each of the LULC classes were developed by reclassifying the output LULC change maps into "no change" in LULC class between two given years and "change" from one LULC class to another LULC class between the same year. A change in LULC class at a given location, for example, LULC class x1 at time 1 to LULC class x2 at time 2, was categorized as a loss in LULC class x1 and a gain in LULC class x2 in the change maps, i.e., a transition from x1 to x2. Based on this method, maps representing losses and gains of the seven aggregated LULC classes in 1975–2000, 2000–2013, and 1975–2013 were generated. The change maps were analyzed for important transitions that could be related to land degradation and recovery of the ecosystem. Transitions from natural LULC classes (e.g., natural vegetation) to human-induced LULC classes (e.g., settlements) were considered as developments that may lead to land degradation, whilst the reverse, i.e., a transition from human-induced LULC classes to natural vegetation, were considered as developments that may lead to land recovery.

The three LULC maps were further cross-tabulated using the "tabulate area" tool in the ARCGIS Software Program (version 10.3.1). The "tabulate area" tool calculates the cross-tabulated areas between two or more datasets and outputs cross-tabulated results of the input data (i.e., "to and from change matrix") [80]. The three maps were analyzed again in terms of loss and gain in area extent over the three periods by calculating the relative net loss/gain of each of the LULC classes in the three time intervals, i.e., 1975–2000, 2000–2013, and 1975–2013. As done earlier for the 24 LULC classes in 1975 and 2013, the change in area extent for each LULC class at each time interval was estimated, and the change results were divided by the initial area extent of each LULC class. The 1975 LULC area extent was taken as the initial reference, and therefore the LULC change in each interval was referred to this value. The output results were each multiplied by hundred to obtain the changes in percentage.

This gave information about the magnitude and the direction of change in the three time intervals and served as information to interpret land degradation or recovery in this region.

Furthermore, Equation (1) below was used to calculate the annual rate of change of each LULC class for each transition period.

$$\frac{A2 - A1}{A1(Y2 - Y1)} \times 100 \tag{1}$$

where $A1$ is the initial area extent of each LULC class for each period at year 1 ($Y1$) and $A2$ is the final area extent of each LULC class at year 2 ($Y2$). Subsequently, the LULC classes, i.e., forestland, cropland, and other vegetation, were merged into one broader vegetation cover class (total vegetation cover) to determine the overall losses and gains. We regarded the extent of vegetation cover as the main indicator of land conditions, i.e., a decrease in the total vegetation cover indicates land degradation. However, losses and gains can occur at the same time at different locations, and so the loss and gain maps from the LULC transition analyses need to be placed side by side to fully understand the replacement of one LULC class by another LULC class. The net relative change in the area covered by the total vegetation cover was calculated by subtracting the overall gross loss in the total vegetation cover from the overall gross gain in two given years. The results were then divided by the entire area of the total vegetation cover in the initial year. The 1975 area extent was used as an initial reference value for each period. The outputs were multiplied by one hundred to get the changes in percentages for the 1975–2000, 2000–2013, and 1975–2013 transition periods.

*3.3. Detailed Hotspots and LULC Transitions Analyses in West Africa and the Two Sub-Regions (Humid and Arid)*

Specific changes in the LULC classes were analyzed for a better understanding of the drivers by looking at which LULC class replaced the initial one in 1975 and 2000. Cross-tabulated statistics in terms of area extent for each map pair in 1975–2000, 2000–2013, and 1975–2013 were calculated [80]. These analyses were done for the seven aggregated LULC classes at three time intervals over the 38 years period. The values (areas) in the diagonal cells of each row of the output tables ("to and from change matrix") are the areas which remained "unchanged", while the values in the off-diagonal elements are the "changed" areas for a given period. The total sum of the areas in the off-diagonal elements of each row of the output table gives the gross loss in area extent of each LULC class during a given period.

Similarly, the total sum of the areas in the off-diagonal elements of each column of the output table gives the gross gain in area extent of each LULC class in a given period. The relative changes (%) by class-pair were calculated as the ratio of the value in each off-diagonal row element in the output table to the area of each class in 1975 or 2000 (i.e., the initial reference years). We scanned the percentage of each transition in the "to and from change matrix" (see Tables 2 and A3) for the whole of West Africa and the two sub-regions (arid and humid) to identify transitions with magnitudes greater than 0%. This threshold of transitions was considered significant because international land use policies such as the REDD+'s Policy [36–38] in the study area aim at zero deforestation and forest degradation. Therefore, all the transitions greater than 0% in magnitude were considered more important in shaping the environmental landscape. Representative samples were selected as "important transitions" to be analyzed in detail at the local level [80].

Widespread changes in the various LULC types were observed on the LULC change maps in 1975–2000, 2000–2013, and 1975–2013. To smooth the LULC changes and reduce the random noise on the change maps, different moving windows of majority filters were explored. A majority filter of 5-by-5-pixel moving window was ideal to remove the random noise on the change maps [76–78]. At this window, the change maps were more or less segmented and fractional abundance of each transition and hotspots were clearly identified. Therefore, in the subsequent analyses, a majority filter of 5-by-5-pixel moving window [79–81] was applied to the 1975–2013 LULC loss and gain maps (see Section 3.2 for detailed description) to create two hotspot maps for the period 1975–2013, i.e., hotspots of LULC loss for 1975–2013 and hotspots of LULC gain for 1975–2013. Sample areas of such

hotspots were extracted from the 1975 and 2013 LULC maps and analyzed in detail at the local level to understand how the output statistics relate to the actual spatial pattern of changes. This enabled us to identify the proximate/drivers of the LULC transitions at each hotspot.

Furthermore, based on the broad bio-climatic zones in West Africa [74], the LULC transitions in the humid (Sudanian, Guinean, and Guineo-Congolian) and the arid–semi-arid (Sahel) regions of West Africa were compared by extracting and analyzing the LULC transitions in terms of loss/gain in area for each sub-region (humid and arid). The underlying driving factors of the LULC transitions we identified from some of the hotspots were reviewed from the literature and categorized as either climate drivers, anthropic drivers, or a combination of the two drivers.

## 4. Results

### 4.1. Changes in the Original Twenty-Four LULC Classes

The extents and the relative net changes (%) in the areas covered by the 24 LULC classes in the original USGS data (see Figures 2 and A2 in the Appendix A) illustrate the positive and negative effects of anthropic activities on the various LULC classes. Due to anthropic influence on the natural environment and exploitation of the natural resources for economic development and livelihood support, all the natural LULC classes, i.e., forestland, savannah, grassland, water bodies, woodland, mangrove, and swamp forest, in the 24 LULC classes except herbaceous and thicket vegetation suffered a net relative reduction during 1975–2013. In contrast, the LULC classes related to exploitation of natural resources, i.e., open-mine fields, settlements, plantations, and irrigated and rain-fed agricultural land expanded significantly during the same time frame. Open-mine fields increased enormously by 512.2% and were observed to be degrading forestland and savannah vegetation. Irrigated agriculture increased by 237.1% and was found to be associated with shrinking wetlands and swamp forest. Plantations for rubber, cashew nut, and oil palm plantations also recorded a substantial gain of 149.0%. In addition, rain-fed agricultural land also increased by 110.3%. Such an encroachment was at the expense of all the different forms of natural vegetation. Settlement expansion (i.e., 139.7%) came at the expense of savannah vegetation, forestland, and cropland.

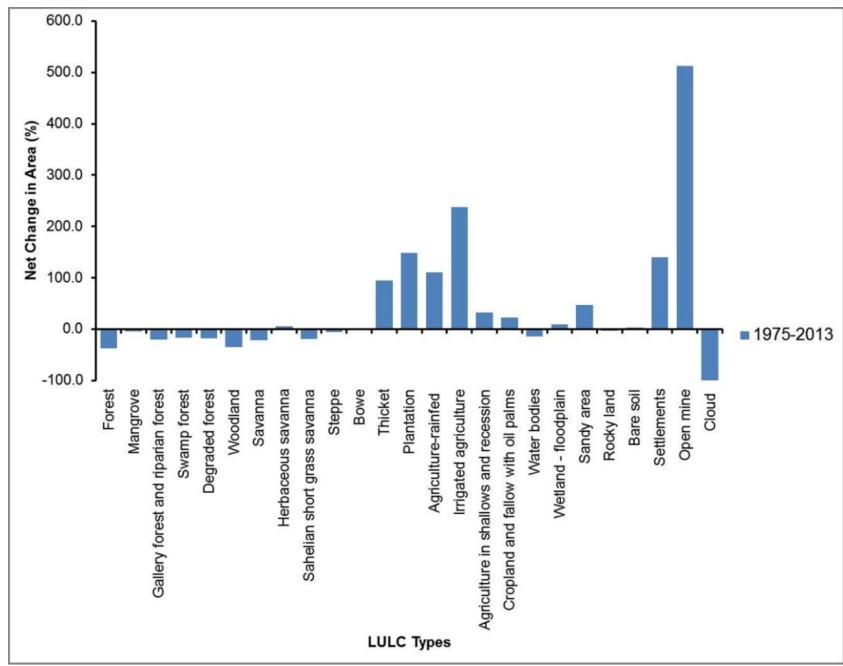

**Figure 2.** The relative net change in area extent of the original USGS 24 Land Use Land Cover (LULC) classes/types from 1975 to 2013.

## 4.2. Fractional Abundance of the Seven Aggregated LULC Classes

The detailed analyses of the LULC changes and transitions were based on the reclassified LULC maps with seven classes (see Figure 3). The maps for 1975, 2000, and 2013 show the seven major LULC categories defined in Table 1. The fractional abundances of the seven aggregated LULC classes in 1975, 2000, and 2013 are shown in Figure 4. The greatest part of the mapped area was covered by "other vegetation", accounting for 68.2%, 62.9%, and 57.6% of the total area in 1975, 2000, and 2013, respectively. Cropland increased significantly from about 11.5% in 1975 to 18% in 2000 and 23.8% in 2013 in the total mapped area. Forestland decreased slightly with fractional abundance of 11.9% in 1975, 9.9% in 2000, and 8.7% in 2013. Settlements increased substantially from 0.3% in 1975 to 0.5% in 2000 and 0.7% in 2013.

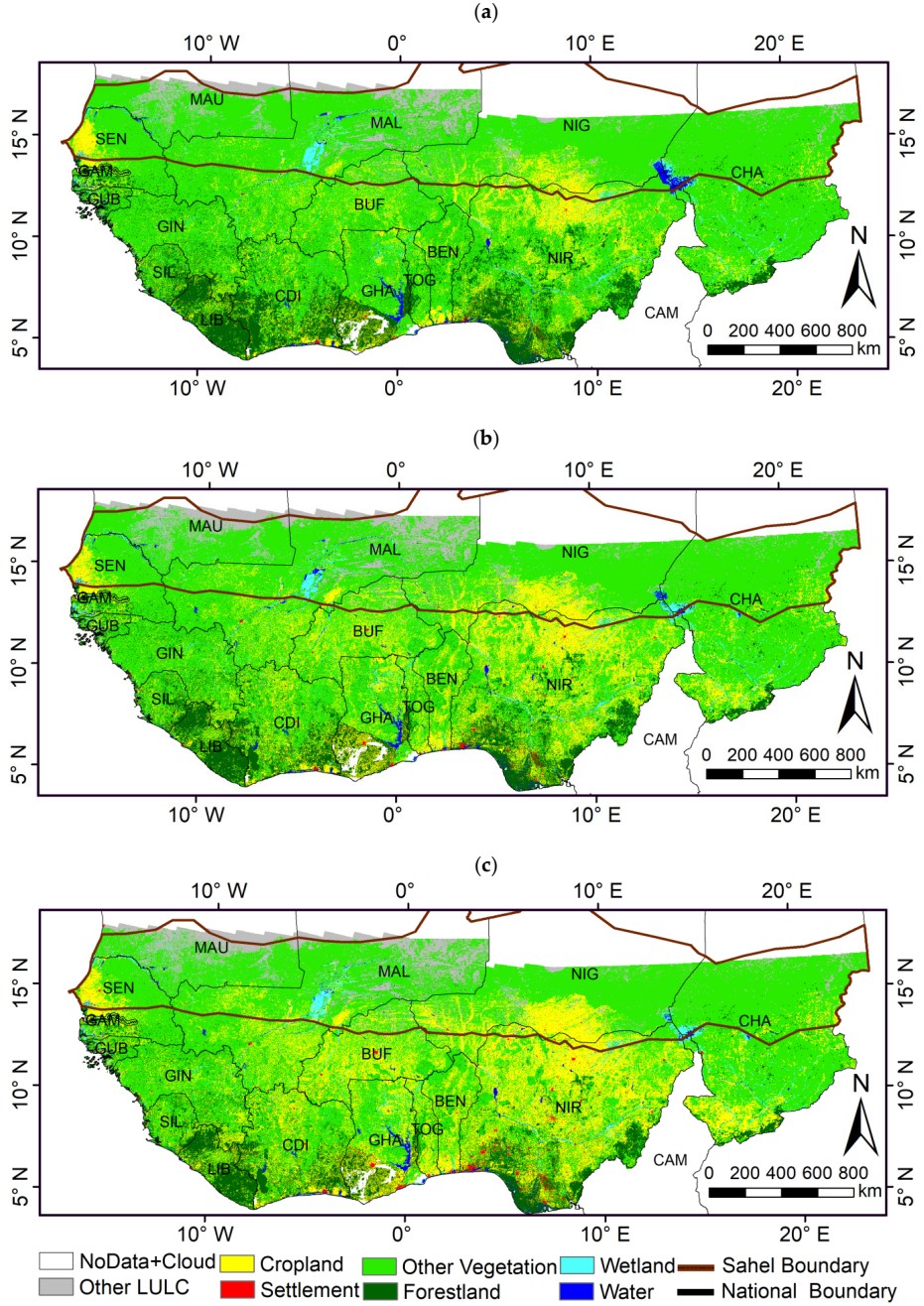

**Figure 3.** The reclassified Land Use Land Cover (LULC) maps of West Africa in: (**a**) 1975; (**b**) 2000; (**c**) 2013.

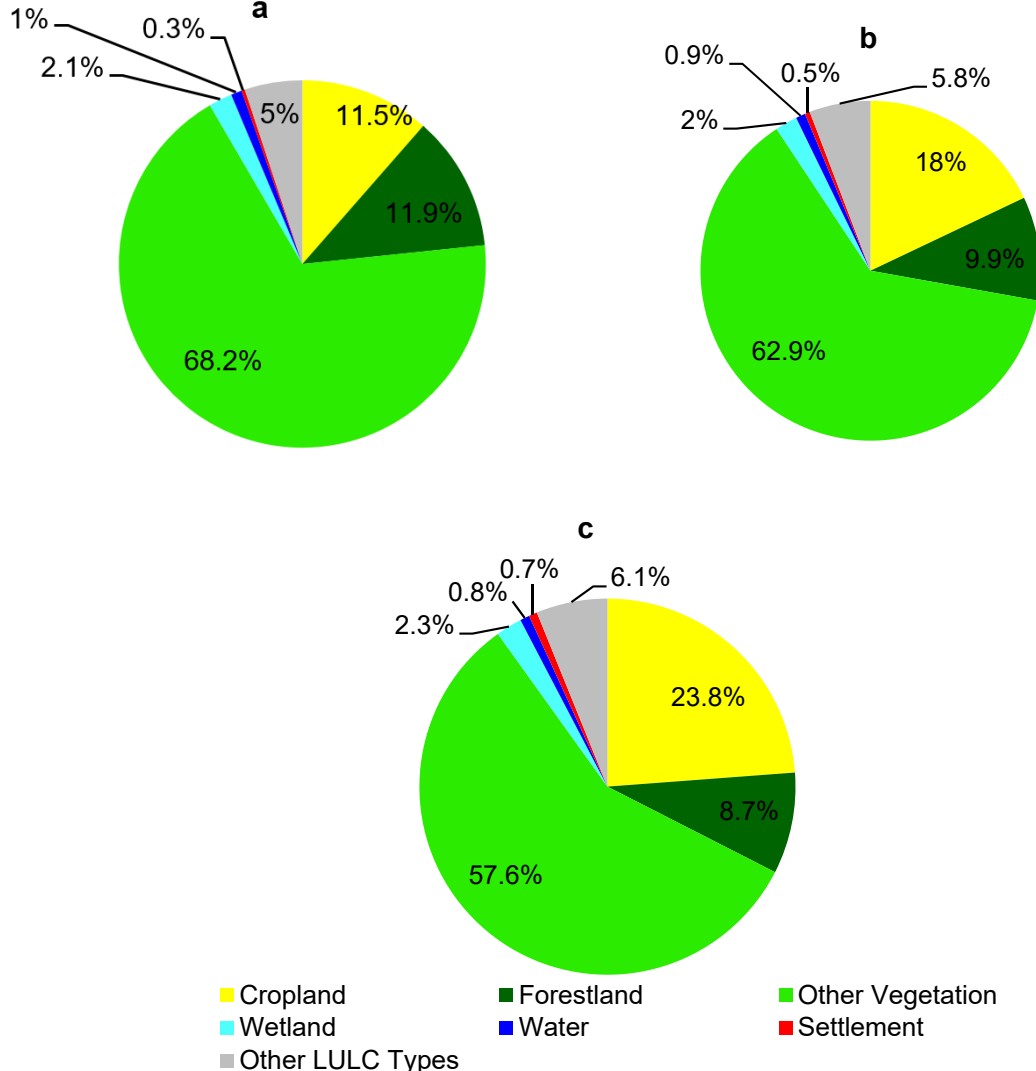

**Figure 4.** Fractional abundance of the seven aggregated Land Use Land Cover (LULC) classes/types in West Africa in: (**a**) 1975; (**b**) 2000; and (**c**) 2013.

*4.3. The General LULC Changes Based on the Reclassified LULC Maps in West Africa and the Two Sub-Regions (Humid and Arid) during 1975–2000, 2000–2013, and 1975–2013*

The results in Figures 5 and 6, Tables 2 and A1–A3, (Figures A3 and A4 in Appendix A, indicate that each of the seven aggregated LULC classes experienced sizeable changes during 1975–2000, 2000–2013, and 1975–2013. Out of the total area, which was approximately 5*10$^6$ km$^2$ of the West African subcontinent mapped in this study, 27.9% of the area changed in 1975–2013, whilst about 72.1% of the area remained unchanged. Such changed (unchanged) areas in 1975–2000 and 2000–2013 were 14.8% (85.2%) and 13.1% (86.9%), respectively. Overall, the LULC transitions (see Figure 5) observed in the semi-arid and the arid (Sahel) as well as the humid (Sudanian, Guinean, Guineo-Congolian) regions of West Africa were not too different from the transitions observed in the whole of West Africa. A detailed summary of the net changes of the various LULC types and the LULC transition matrix in the whole of West Africa and the humid and arid regions are presented in Tables 2 and A1–A3, (Appendix A).

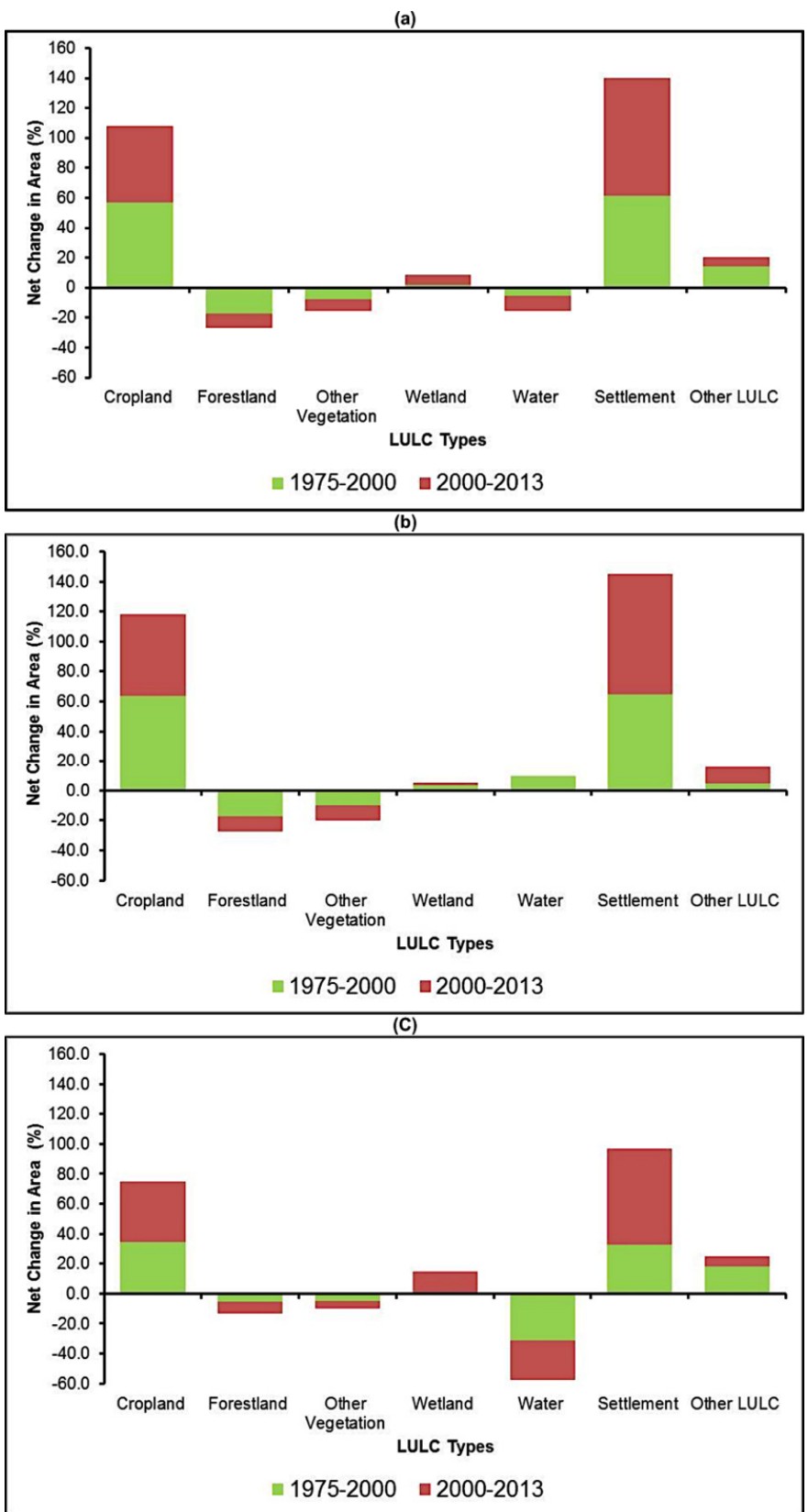

**Figure 5.** The relative net changes (%) of the seven aggregated Land Use Land Cover (LULC) classes/types during 1975–2000 and 2000–2013 in the whole of West Africa (**a**) and the two sub-regions, humid (**b**) and arid (**c**), in West Africa. The sum of the losses and gains for the two periods is the relative net changes in 1975–2013.

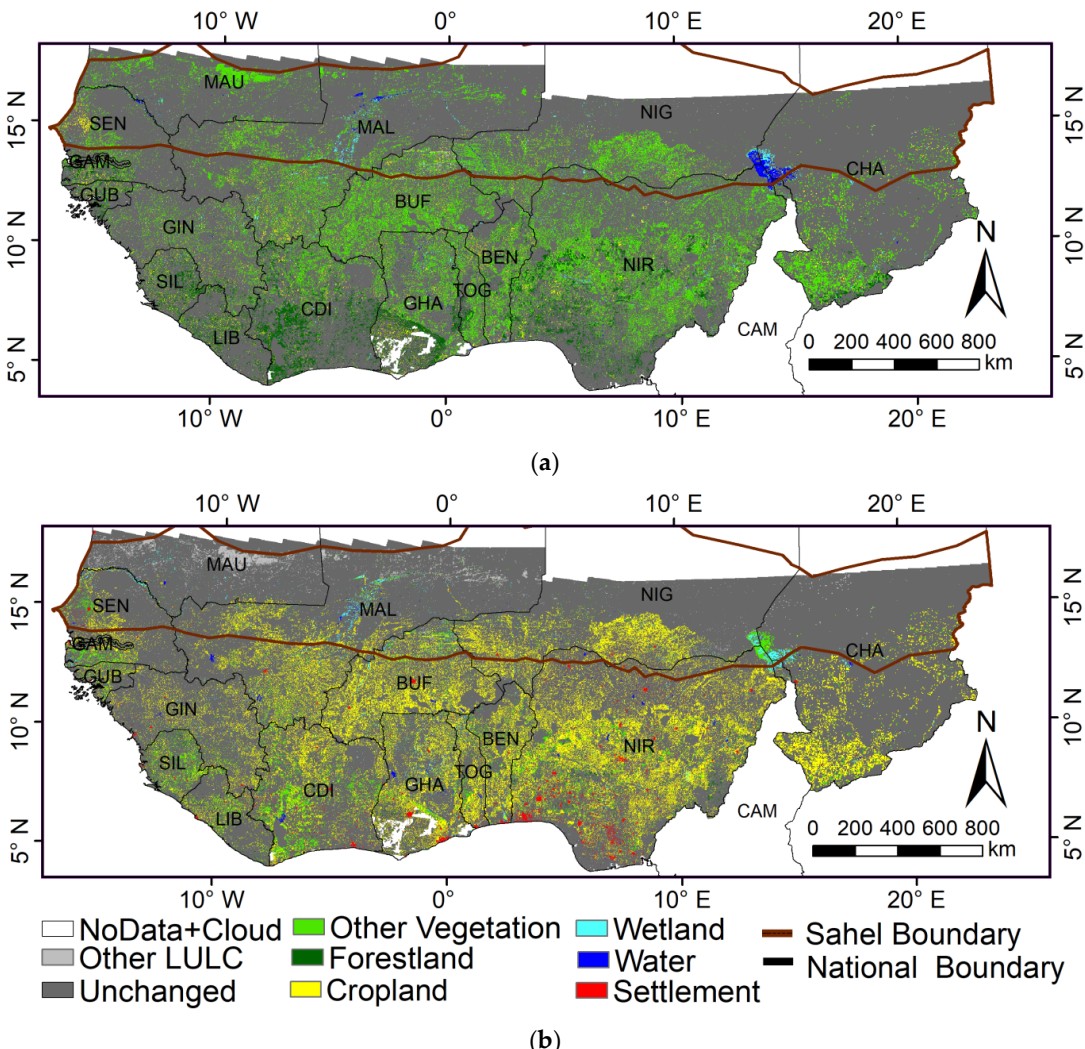

**Figure 6.** Land Use Land Cover (LULC) loss (**a**) and gain (**b**) maps of West Africa between 1975 and 2013; the legend indicates that the LULC classes/types disappeared in 2013 in (**a**), and new LULC classes/types appeared in 2013 in (**b**).

**Table 2.** The Land Use Land Cover (LULC) transition matrix of the seven aggregated LULC classes/types during 1975–2000, 2000–2013, and 1975–2013 transitions periods in West Africa; the relative changes by class (%) are calculated by dividing the area in each off-diagonal element by the area of the class indicated in the left-most column (in Table A1 in the Appendix A) for the initial reference year of each period, i.e., either 1975 for the periods (1975–2000) and (1975–2013) or 2000 for the period (2000–2013). The percentage change in each line (moving from left to right) indicates the percentage loss in a given LULC type as a result of a transition to a different LULC class at a given period. Each interval/period of the LULC transitions is highlighted in "grey".

| LULC Type | Cropland | | Forestland | | Other Vegetation | | Wetland | | Water | | Settlement | | Other LULC | |
|---|---|---|---|---|---|---|---|---|---|---|---|---|---|---|
| | Area (km²) | % | Area (km²) | % | Area (km²) | % | Area (km²) | % | Area (km²) | % | Area (km²) | % | Area (km²) | % |
| **Period (1975–2000)** | | | | | | | | | | | | | | |
| Cropland | 301,576 | 53.9 | 41,488 | 7.4 | 191,012 | 34.1 | 10,284 | 1.8 | 3764 | 0.7 | 8064.0 | 1.4 | 3528 | 0.6 |
| Forestland | 91,048 | 15.6 | 293,616 | 50.5 | 180,824 | 31.1 | 4220 | 0.7 | 5192 | 0.9 | 5148 | 0.9 | 1896 | 0.3 |
| Other Vegetation | 455,400 | 13.7 | 135,404 | 4.1 | 2,576,824 | 77.3 | 42,688 | 1.3 | 12,180 | 0.4 | 6552 | 0.2 | 105,324 | 3.2 |
| Wetland | 15,216 | 15.0 | 2968 | 2.9 | 39,396 | 38.9 | 34,860 | 34.5 | 5636 | 5.6 | 344 | 0.3 | 2756 | 2.7 |
| Water | 3648 | 7.9 | 4480 | 9.6 | 12,792 | 27.5 | 8208 | 17.7 | 16,384 | 35.3 | 356 | 0.8 | 596 | 1.3 |
| Settlement | 5576 | 37.5 | 2520 | 17.0 | 2764 | 18.6 | 300 | 2.0 | 268 | 1.8 | 3348 | 22.5 | 84 | 0.6 |
| Other LULC | 4932 | 2.0 | 1492 | 0.6 | 69,008 | 27.8 | 2408 | 1.0 | 456 | 0.2 | 140 | 0.1 | 169,700 | 68.4 |
| Total | 877,396 | | 481,968 | | 3,072,620 | | 102,968 | | 43,880 | | 23,952 | | 283,884 | |
| **Period (2000–2013)** | | | | | | | | | | | | | | |
| Cropland | 805,824 | 91.8 | 6636 | 0.8 | 57,104 | 6.5 | 1808 | 0.2 | 204 | 0.0 | 5480 | 0.6 | 340 | 0.0 |
| Forestland | 33,756 | 7.0 | 411,968 | 85.5 | 32,324 | 6.7 | 1264 | 0.3 | 388 | 0.1 | 2060 | 0.4 | 208 | 0.0 |
| Other Vegetation | 314,496 | 10.2 | 6176 | 0.2 | 2,717,000 | 88.4 | 11,464 | 0.4 | 1948 | 0.1 | 4416 | 0.1 | 17,120 | 0.6 |
| Wetland | 6928 | 6.7 | 356 | 0.3 | 4528 | 4.4 | 89,092 | 86.5 | 1812 | 1.8 | 160 | 0.2 | 92 | 0.1 |
| Water | 1444 | 3.3 | 60 | 0.1 | 1520 | 3.5 | 6036 | 13.8 | 34,716 | 79.1 | 24 | 0.1 | 80 | 0.2 |
| Settlement | 408 | 1.7 | 52 | 0.2 | 64 | 0.3 | 0 | 0.0 | 12 | 0.1 | 23,404 | 97.7 | 12 | 0.1 |
| Other LULC | 500 | 0.2 | 12 | 0.0 | 1612 | 0.6 | 236 | 0.1 | 80 | 0.0 | 120 | 0.0 | 281,324 | 99.1 |
| Total | 1,163,356 | | 425,260 | | 2,814,152 | | 109,900 | | 39,160 | | 35,664 | | 299,176 | |
| **Period (1975–2013)** | | | | | | | | | | | | | | |
| Cropland | 339,356 | 60.6 | 36,712 | 6.6 | 154,560 | 27.6 | 10,452 | 1.9 | 3504 | 0.6 | 11,536 | 2.1 | 3596 | 0.6 |
| Forestland | 125,704 | 21.6 | 259,484 | 44.6 | 177,008 | 30.4 | 4224 | 0.7 | 5404 | 0.9 | 8056 | 1.4 | 2064 | 0.4 |
| Other Vegetation | 659,008 | 19.8 | 118,788 | 3.6 | 2,367,624 | 71.0 | 47,740 | 1.4 | 11,152 | 0.3 | 10,680 | 0.3 | 119,380 | 3.6 |
| Wetland | 21,464 | 21.2 | 2680 | 2.6 | 34,984 | 34.6 | 34,740 | 34.3 | 3900 | 3.9 | 564 | 0.6 | 2844 | 2.8 |
| Water | 5104 | 11.0 | 4120 | 8.9 | 11,856 | 25.5 | 9784 | 21.1 | 14,528 | 31.3 | 492 | 1.1 | 580 | 1.2 |
| Settlement | 5708 | 38.4 | 2116 | 14.2 | 2300 | 15.5 | 272 | 1.8 | 276 | 1.9 | 4084 | 27.5 | 104 | 0.7 |
| Other LULC | 7012 | 2.8 | 1360 | 0.5 | 65,820 | 26.5 | 2688 | 1.1 | 396 | 0.2 | 252 | 0.1 | 170,608 | 68.8 |
| Total | 1,163,356 | | 425,260 | | 2,814,152 | | 109,900 | | 39,160 | | 35,664 | | 299,176 | |

*4.4. The Major LULC Transitions Observed in West Africa and the Two Sub-Regions (Humid and Arid) during 1975–2000, 2000–2013, and 1975–2013*

The major changes we observed for the entire periods of the analyses were gains in settlement, cropland, wetland, and other anthropic LULC types and losses in forestland, other vegetation, and water bodies (see Figure 5 and Table A1 in the Appendix A). The key transitions were the transitions of forestland and "other vegetation" into settlement and cropland (see Table 2). The reduction of forestland in West Africa was rather large, i.e., −17.2%, −9.7%, and −26.9% in 1975–2000, 2000–2013, and 1975–2013, respectively. The annual rate of change (−0.9%) in forestland was higher in the period of 2000–2013 (i.e., the post-drought era and recent period) than the annual rate of change (−0.7%) in 1975–2000 (i.e., the drought era and soon after the drought, see Table A1 in the Appendix A). The same trend (see Table A2 in the Appendix A) was detected for forestland losses in the two sub-regions (humid and arid). Other vegetation constituted the highest fraction of the total vegetation cover in the study area, yet experienced the largest loss. This is because other vegetation types, e.g., shrubland, grassland, and so forth, are highly impacted by a combination of climate and anthropic drivers. Other vegetation cover decreased by −7.9%, −7.8%, and −15.6% in 1975–2000, 2000–2013, and 1975–2013, respectively, in the whole of West Africa. The annual rate of change (−0.6%) in "other vegetation" was higher in the period of 2000–2013 (i.e., the post-drought era and recent period) as compared to the annual rate of change, i.e., −0.3% and −0.4% recorded in 1975–2000 and 1975–2013, respectively (see Table A1 in the Appendix A). This trend was similar in the two sub-regions (humid and arid) in West Africa (see Table A2 in the Appendix A).

The relative net losses (%) in the total vegetation cover, i.e., other vegetation, forestland, and cropland, in West Africa were −0.98%, −0.7%, and −1.60% in 1975–2000, 2000–2013, and 1975–2013, respectively. These relative net losses would have been higher, i.e., −9.2%, −8.1%, and −17.3% in 1975–2000, 2000–2013, and 1975–2013, respectively, without the gains in cropland. In the humid region, the relative net losses (%) in total vegetation cover, i.e., other vegetation, forestland, and cropland combined were −0.5%, −0.4%, and −0.9% in 1975–2000, 2000–2013, and 1975–2013, respectively, whilst in the arid region, the relative net losses (%) in total vegetation cover were −1.9%, −1.0%, and −2.9% in 1975–2000, 2000–2013, and 1975–2013, respectively. Shrinking of open water bodies followed the same trend as the losses in forestland and other vegetation in the whole of West Africa and the arid region. In contrast, relative net gain in open water bodies was observed in the humid region during the period of 1975–2000 with a slight loss in 2000–2013.

A detailed pixel-by-pixel comparison of the new and old LULC types for each observed LULC change in the form of a transition matrix for the periods of 1975–2000, 2000–2013, and 1975–2013 was undertaken for the whole of West Africa (Table 2) and the two sub-regions (humid and arid, see Table A3 in the Appendix A). The centroids of some hotspots of the LULC transitions identified in the maps between 1975 and 2013 are shown in Figure 7. These hotspots were zoomed-in on the 1975 and 2013 reclassified maps to produce detailed maps of some of the local hotspots (Figures 8–12) for subsequent underlying driving factors analyses based on a review of previous literature about these hotspots. We then categorized the identified underlying driving factors as either climate, anthropic, or a combination of the two. Full details of the major LULC transitions we identified at some hotspots are presented in the subsequent sections.

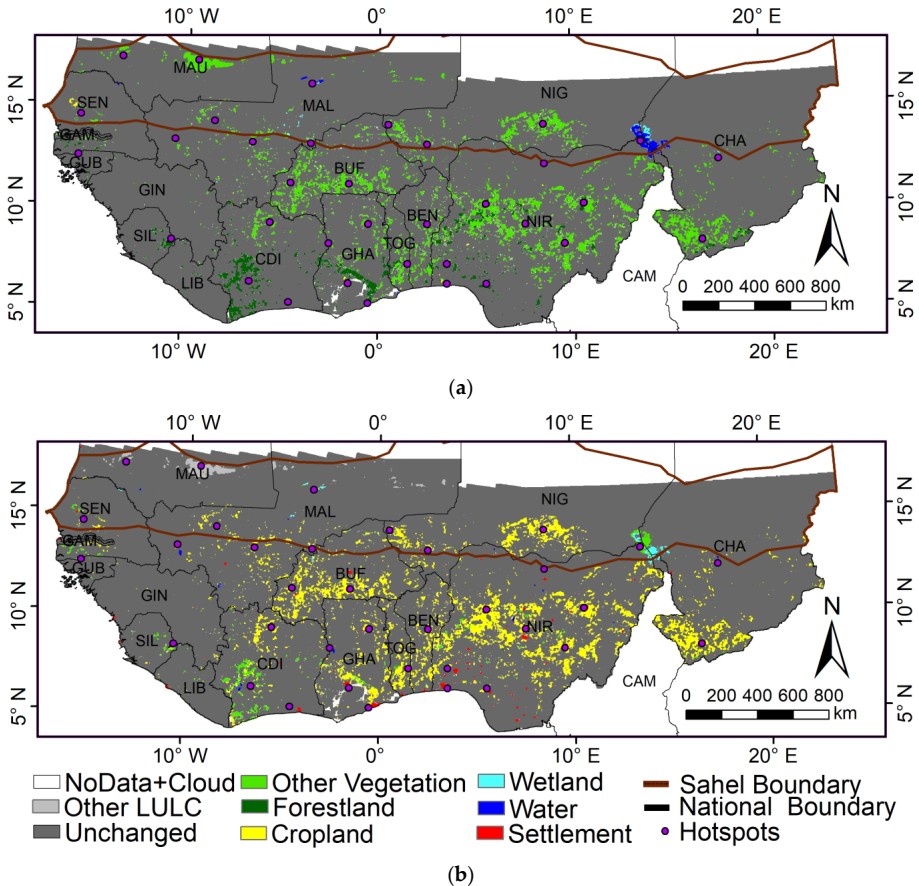

**Figure 7.** Hotspots of Land Use Land Cover (LULC) loss and gain in the period of 1975–2013. The legend indicates that the LULC classes/types disappeared in 2013 in (**a**), and new LULC classes appeared in 2013 in (**b**). The purple dots are centroids of some important hotspots of the LULC change identified by this study.

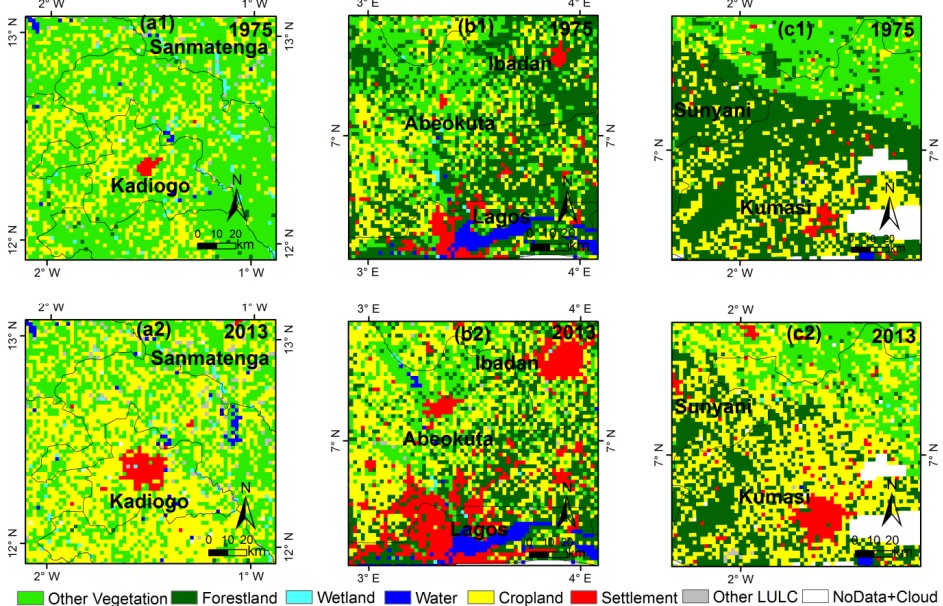

**Figure 8.** Transition of other vegetation to settlement between 1975 (**a1**) and 2013 (**a2**) around Kadiogo in Burkina Faso; transition of forestland to settlement between 1975 (**b1**) and 2013 (**b2**) around Ibadan and Lagos in Nigeria and transition of cropland to settlement between 1975 (**c1**) and 2013 (**c2**) around Kumasi in Ghana.

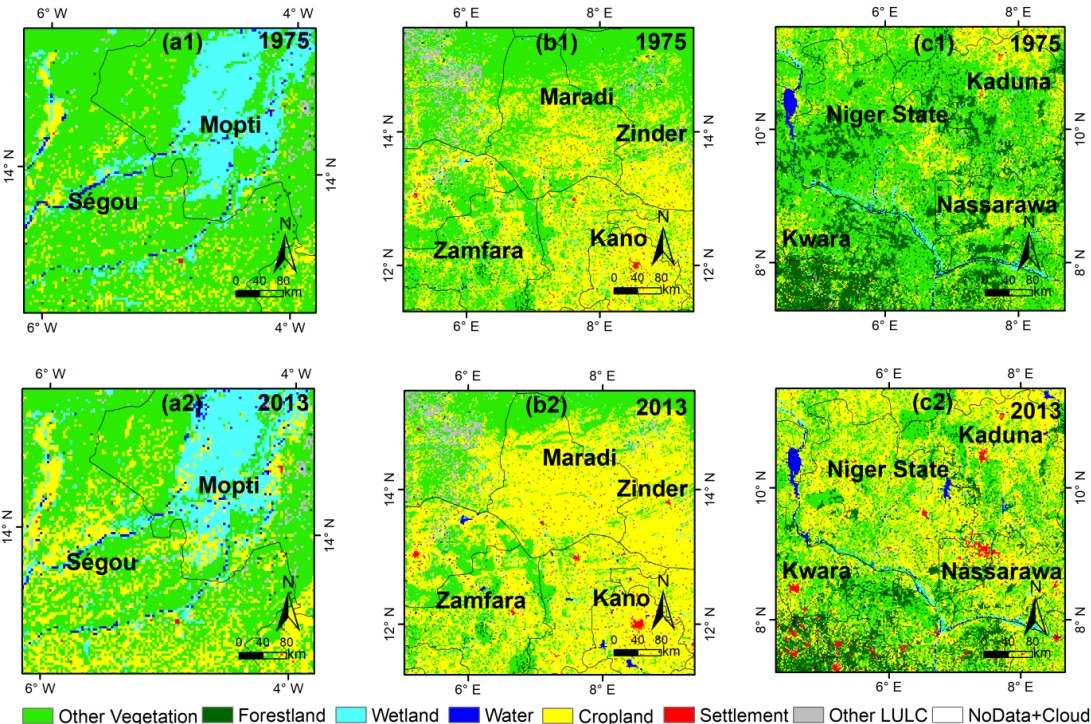

**Figure 9.** Transition of wetland to cropland between 1975 (**a1**) and 2013 (**a2**) around Segou and Mopti in Mali; transition of other vegetation to cropland between 1975 (**b1**) and 2013 (**b2**) around Zinder and Maradi in Niger as well as Zamfara and Kano in Nigeria; transition of forestland to cropland between 1975 (**c1**) and 2013 (**c2**) around Niger State, Kwara and Nassarawa in Nigeria.

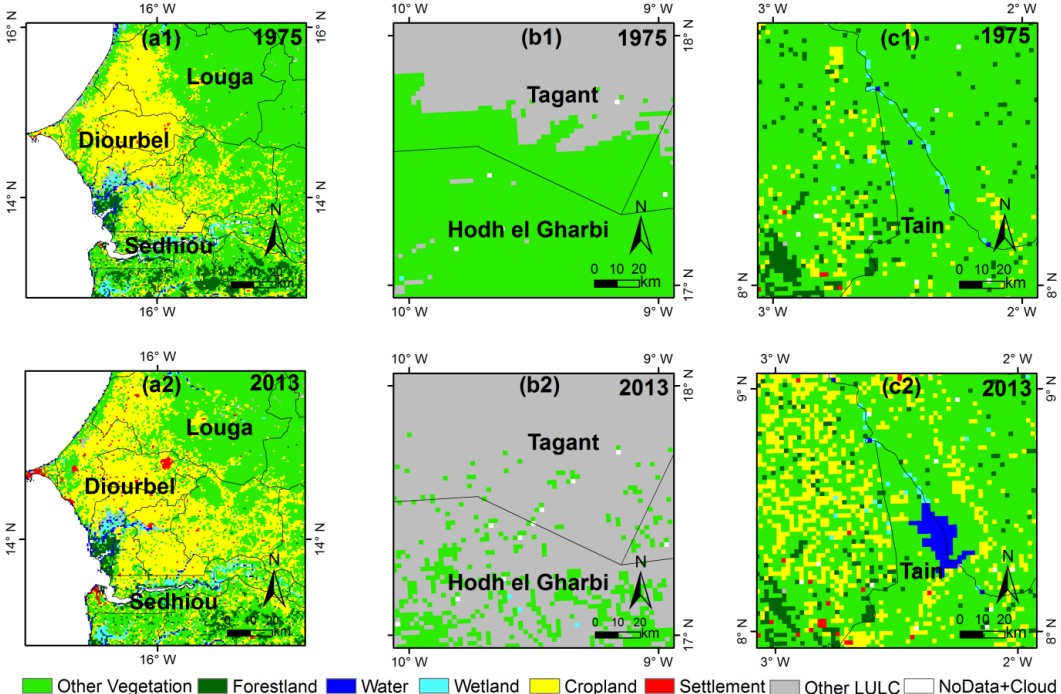

**Figure 10.** Transition of cropland to other vegetation and settlement between 1975 (**a1**) and 2013 (**a2**) around Louga and Diourbel in Senegal; transition of other vegetation to other LULC types between 1975 (**b1**) and 2013 (**b2**) around Tagant and Hodh el Gharbi in Mauritania; and transition of other vegetation to water bodies between 1975 (**c1**) and 2013 (**c2**) around Tain District in Ghana.

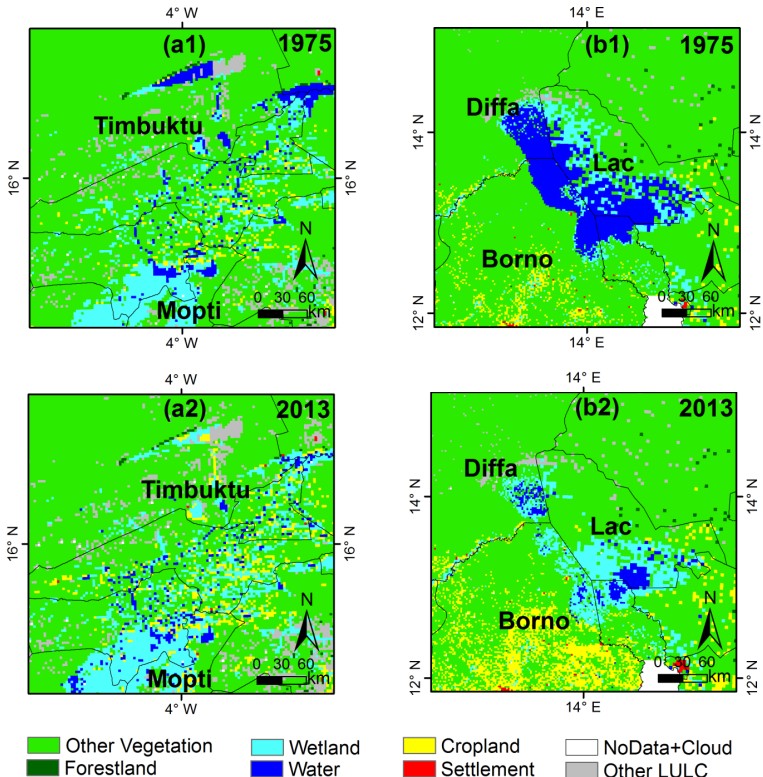

**Figure 11.** Transition of water bodies to wetland between 1975 (**a1**) and 2013 (**a2**) around Timbuktu in Mali and transition of water bodies to other vegetation and wetlands between 1975 (**b1**) and 2013 (**b2**) around Lac in Chad, Borno in Nigeria and Diffa in Niger.

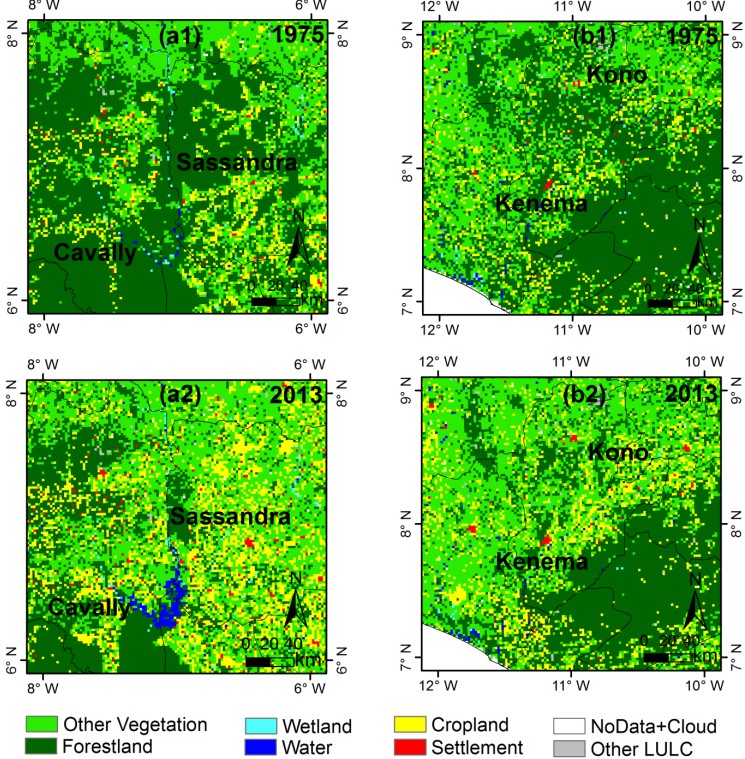

**Figure 12.** Transition of forestland to water bodies between 1975 (**a1**) and 2013 (**a2**) around Sassandra and Cavally in Cote d'Ivoire and transition of forestland to other vegetation between 1975 (**b1**) and 2013 (**b2**) around Kono and Kenema in Sierra Leone.

### 4.4.1. Settlement Expansions

Despite the fact that the fractional abundance of the total area covered by settlement in the final year (2013) of the analyses was less than 1% (see Figure 4) in the entire study area, settlement expansions accounted for the largest transition in terms of relative net changes in area (see Figure 5 and Table A1 in the Appendix A). Figure A4 in the Appendix A is the extent of the seven aggregated LULC classes during 1975, 2000, and 2013 in West Africa. During the period of 1975–2000 (i.e., the drought era and soon after the drought), the relative net gain in settlement was 61.2%, while it was 140% over the 38-year period (1975–2013, i.e., more than doubled). Settlement increased by 79% during the period of 2000–2013 (i.e., the post-drought era and recent period), which is higher than the amount recorded in the drought era and soon after the drought. In the period of 1975–2000 (i.e., the drought era and soon after the drought), the relative net gain (64.6%) in settlements in the humid region was twice the relative net gain (32.6%) in the arid and semi–arid regions during the same period (see Figure 5b,c and Table A2 in the Appendix A). The annual rate of change in the humid region during this period was 2.6% per year, whilst in the arid region; it was 1.3% per year (see Table A2 in the Appendix A).

On the other hand, in 1975–2013 (i.e., the entire period), the relative net gain (145.3%) in settlement in the humid region was about 1.5 times the relative net gain (96.6%) in the arid region (see Table A2 in the Appendix A). During the period of 2000–2013 (i.e., the post-drought era and recent period), settlement expansion in both the arid and the humid regions followed a similar pattern, with the net gain (64.0%) in settlements in the arid region being comparable with the relative net gain (80.6%) in the humid region. The annual rate of change in the humid region was 3.8% per year and 3.7% per year in the arid region (see Table A2 in the Appendix A). Figure 5 is a comparison of the relative net changes in the various LULC classes in the whole of West Africa and the two sub-regions (humid and arid).

Hotspots of settlement gain/expansion can be inferred from Figure 7b. Predominantly, this transition was detected in the middle belt and the coastal regions of the southern part of West Africa and more evident in areas such as Lagos, Ibadan, and Abuja in Nigeria; Bobo-Dioulasso, Ougadougou and Kadiogo region in Burkina Faso; as well as Kumasi and Accra in Ghana and many other places such as Abidjan in Cote D'Ivoire, Dakar in Senegal and Bamako in Mali.

Expansion of settlements at the expense of other vegetation, cropland, and forestland constituted one of the largest transitions in all the three intervals of our analyses in the whole of West Africa. In other words, settlement expansion is one of the major drivers of forestland, other vegetation and cropland losses in West Africa. In the humid region, settlement expansion was at the expense of other vegetation, cropland and forestland, whilst in the arid region, settlement expansion was mainly at the expense of other vegetation and other LULC types (see Table A3 in the Appendix A).

Part of the settlement expansion may be explained by a conversion of a fraction of other vegetation cover in 1975 to settlements in 2013. More precisely, the transitions of other vegetation cover to settlements were 0.2%, 0.1%, and 0.3% in 1975–2000, 2000–2013, and 1975–2013, respectively (see Table 2). For example, around Kadiogo in Burkina Faso, other vegetation in 1975 (see Figure 8a1) transitioned into settlements in 2013 (see Figure 8a2). Apart from the settlement gain at the expense of other vegetation in this area (see Figure 8a1,a2), open water bodies' gains at the expense of wetland and other vegetation alongside cropland expansion at the expense of some pockets of forestland and other vegetation are also observable.

Furthermore, a considerable percentage of the total settlement gains in the study area were at the expense of forestland. The transitions of forestland to settlements were 0.9%, 0.4%, and 1.4% in 1975–2000, 2000–2013, and 1975–2013, respectively (see Table 2). This kind of LULC conversion is a clear indication of the impact of anthropic activities on the ecosystem and was more evident in the southern part of West Africa, with the surroundings of Ibadan and Lagos in Nigeria being good examples (see Figure 8b1,b2).

As mentioned before, settlement gain at the expense of cropland was also extensive. The transitions of cropland to settlement were 1.4%, 0.6%, and 2.1% in 1975–2000, 2000–2013, and 1975–2013, respectively (see Table 2). These conversions were common in peri-urban areas in the Ashanti Region of Ghana,

where locations in some cities such as Nkawie, Offinso and Ejusi under intensive crop cultivation in 1975 (see Figure 8c1) were converted to settlements by 2013 (see Figure 8c2) due to their proximity to the central business capital Kumasi. The areas around Kaduna and Nassarawa in Nigeria are other remarkable examples of this LULC conversion. In these areas, settlements expanded considerably between 1975 and 2013 at the expense of cropland (see Figure 9c1,c2). More details of the percentage of settlement gain from each of the LULC types are shown in Table 2.

4.4.2. Cropland Expansions

Another transition worth mentioning in the West African subcontinent was the massive expansion of cropland with a relative net gain of 56.8% in 1975–2000 and 51.1% during the 2000–2013 periods, adding up to 107.8% by 2013. Cropland expansion was larger in the southern, more humid, areas of West Africa, where at the same time the expansion of human settlements took place. The LULC class "other vegetation" (Figure 6a) in 1975 was mostly replaced by cropland (Figure 6b) in 2013. This LULC transition was more widespread in areas such as Nigeria (Niger State), Burkina Faso (Ziro, Nahouri and Sanguie), Benin (Ogou and Tchamba), Ghana (Sisssala and Nadowli), Chad (Lanya and Kabbia), Niger (Zinder, Maradi and Tiliberi), and many other areas.

Although massive expansions of cropland were observed in both the humid and arid regions of West Africa during 1975–2000 (i.e., the drought era and soon after the drought) and 1975–2013 (entire period), the relative net gains (63.7% and 118.0%) in cropland in the humid region were about twice the relative net gains (34.2% and 74.7%) in the arid region, respectively (see Figure 5 and Table A3 in the Appendix A). On the other hand, in 2000–2013 (i.e., the post-drought era and recent period), the relative net gain (54.4%) in cropland in the humid region was not significantly different from the relative net gain (40.5%) in the arid region during the same period (see Figure 5 and Table A2 in the Appendix A). This implies that in both the humid and the arid regions, croplands have been expanding at the same pace in recent years, i.e., 2000–2013, than in the earlier years of 1975–2000. Annual rates of changes for cropland were 2.5%, 2.6%, and 3.1% per year in 1975–2000, 2000–2013, and 1975–2013 in the humid region against the annual rates of changes 1.4%, 2.3%, and 2.0% per year in the arid region during the same periods, respectively (see Table A2 in the Appendix A).

Most of the gains in cropland observed in the West African subcontinent were at the expense of other vegetation, forestland, and wetlands (see Table 2). In the humid regions, cropland expansions were mostly at the expense of forestland, other vegetation, and wetland, while cropland expansions in the arid regions were mainly at the expense of other vegetation (see Table A3). In other words, cropland expansion is also one of the major driving factors of forestland, other vegetation, and wetland losses in West Africa.

In the entire West African subcontinent, appreciable percentages, i.e., 15.0%, 6.7%, and 21.2%, of the initial wetland areas transitioned into cropland in 1975–2000, 2000–2013, and 1975–2013, respectively (see Table 2). Transitions of wetlands to cropland could be explained by conversion of wetlands to irrigated cropland. In the 1975, 2000, and 2013 original LULC maps with 24 classes (see Figure A1 in the Appendix A), such transitions were identified at locations where expansions of irrigated croplands were observed. Clear examples of such locations are Mopti and Segou in Mali around the Inland Niger Delta, where some portions of the areas covered by wetland in 1975 (Figure 9a1) were replaced by cropland in 2013 (Figure 9a2). The transition of wetland to cropland around Lake Chad, Lake Fitiri, and Ouadda in Chad were also noticed. One intriguing aspect of this transition was the wetland loss recorded in the arid region between 1975 and 2000. This may be explained by the impact of the 1970s and 1980s drought on wetlands in this sub-region, but our in-depth analysis of the LULC transitions also revealed that a large part of this wetland loss was due to encroachment by water bodies. The latter suggests human influence on the transition process, e.g., conversion of wetlands to water reservoirs by constructions of dams.

Another significant transition we detected was other vegetation to cropland. This transition was the main LULC transition we observed during the three periods of the analyses. Additionally,

this transition was widespread in the entire study area and explained the largest part of the expansions of cropland. About 13.7%, 10.2%, and 19.8% of the initial other vegetation areas transitioned into croplands in 1975–2000, 2000–2013, and 1975–2013 respectively (see Table 2). This is clearly illustrated by the full–resolution maps (see Figure 9) of some hotspots in Niger, i.e., Zinder and Maradi, and Nigeria, i.e., Zamfara and Kano. At both locations, a large fraction of "other vegetation" areas in 1975 (Figure 9b1) was replaced by "cropland" in 2013 (Figure 9b2).

During the same period of time, transition of forestland to cropland in the West African subcontinent were 15.6%, 7.0%, and 21.6% of the initial forestland areas in 1975–2000, 2000–2013, and 1975–2013, respectively (see Table 2). Typical examples of transition of forestland in 1975 (Figure 9c1) to cropland by 2013 (Figure 9c2) are shown in areas such as Kaduna, Kwara, Nassarawa, and Niger States in Nigeria. These areas correspond to the megacities at the central part of Nigeria where we detected settlement expansions at the expense of natural vegetation. Transition of forestland to cropland was also evident in the hotspot maps around Manya Krobo and Kwahu in Ghana as well as Montagnes in Cote D'Ivoire (see Figure 7).

### 4.4.3. Isolated Cropland Loss

Despite the massive loss in other vegetation cover to cropland (Figure 9b1,b2) over the period of our analyses, the reverse, i.e., the transition of cropland to other vegetation was not negligible. Namely, 34.1%, 6.5%, and 27.6% of the initial cropland areas transitioned to other vegetation in 1975–2000, 2000–2013, and 1975–2013, respectively (see Table 2). This transition was evident in the peanut basin around the city of Diourbel and Louga in Senegal (see Figure 10a1,a2), where a large increase in the other vegetation area at the expense of cropland was observed.

### 4.4.4. Other LULC Types Expansions

The coverage of "other LULC types", e.g., bare land, soil, and open-mine fields, increased by 14.4%, 6.2%, and 20.6% in 1975–2000, 2000–2013, and 1975–2013, respectively, in the whole of West Africa. We discovered that in the arid region, the relative net gains, i.e., 18% and 25.1% in "other LULC types", were higher than in the humid region, i.e., 4.9% and 15.7% in 1975–2000 (i.e., the drought era and soon after the drought) and 1975–2013 (entire period), respectively. However, the relative net gain (10.0%) in "other LULC types" in the humid region in 2000–2013 (i.e., the post-drought era and recent period) was higher than in the arid region, i.e., 7.1% (see Figure 5 and Table A2 in the Appendix A). This may be due to the new open-mine fields, one of the sub-categories of "other LULC types" in our LULC classes, which were more prevalent in the humid region in 2000–2013 (see Figures A1 and A2 in the Appendix A). In the humid region, the relative net gains in "other LULC types" were due to encroachment on "other vegetation", forestland and cropland. In the arid region, the relative net gains in "other LULC types" during these periods were mainly due to encroachment on other vegetation. These translated into 6.8%, 1.1%, and 7.7% of the initial extent of other vegetation in 1975–2000, 2000–2013, and 1975–2013 respectively (see Table A3 in the Appendix A).

About 3.2%, 0.6%, and 3.6% of the areas initially covered by other vegetation, i.e., savannah, steppe, bowe, thicket, herbaceous vegetation, and Sahelian short grasses, in 1975 and 2000 were replaced by "other LULC types", such as sand dunes, bare rocks, open-mine fields, and bare soil in 1975–2000, 2000–2013, and 1975–2013, respectively (see Table 2) in the whole of West Africa. A common example of this transition was discovered around Tagant, Hodh el Gharbi, in Mauritania (see Figure 10b1,b2). It is clear from the two maps, i.e., Figure 10b1,b2, that vast majority of other vegetation areas in 1975 were replaced by "other LULC types" in 2013.

The transition of other vegetation to "other LULC types" detected by this research was almost the same during 1975–2000 and 2000–2013 in the whole of West Africa. However, the transition was different at the two sub-regions (arid and humid). In the arid regions, the transition of other vegetation to other "LULC types" during 1975–2000 (i.e., the drought era and soon after the drought) was higher than 2000–2013 (i.e., the post-drought era and the recent period; see Table A3 in the Appendix A).

The gains in "other LULC types" in the humid region were larger in the periods of 2000–2013 than in 1970–2000. These transitions were more pronounced around locations previously covered with degraded forestland in our study (see Table A3 in the Appendix A).

### 4.4.5. Shrinking of Water Bodies

Water bodies suffered relative net losses during all the periods of our analyses in the whole of West Africa. Water bodies' losses can be noticed in both the loss (see Figure 6a) and gain (see Figure 6b) maps around Chad Lake, the Inland Niger Delta in Mali, and Senegal Basin. In these areas, water bodies in Figure 6a in (1975) were replaced by wetland and other vegetation (Figure 6b) in (2013). The annual rate, i.e., −0.8% of water bodies' loss per year, was higher in 2000–2013 (i.e., the post-drought era and recent period) than in 1975–2000, which was −0.2% per year in the drought era and soon after the severe drought. The annual rate of change in 1975–2013 was −0.4% per year (see Table A1 in the Appendix A). This explains why the relative net loss, i.e., −5.6% in water bodies' extent during 1975–2000, the severe drought era and soon after the drought, was lower than the relative net loss, i.e., −10.2% and −15.7% in 2000–2013 (i.e., the post-drought era and recent period) and in 1975–2013 (entire period), respectively (see Table A1 in the Appendix A). It is interesting to note that the −10.2% loss in 2000–2013 was substantial, although the drought years were much earlier, in the 70s and 80s.

Replacement of water bodies by wetlands explains much of the water bodies' losses in West Africa. Appreciable percentages of the initial areas of water bodies, i.e., 17.7%, 13.8%, and 21.1% in West Africa, transitioned into wetlands in 1975–2000, 2000–2013, and 1975–2013, respectively (see Table 2). The relative net gains in wetland areas during 1975–2000, 2000–2013, and 1975–2013 were 1.8%, 6.9% and 8.6%, respectively. The annual rate of wetland gains during 2000–2013 was 0.5% per year, which was rather large as compared with the annual rate of wetland gains, i.e., 0.1% and 0.2% per year in 1975–2000 and 1975–2013, respectively (see Table 2). This is partly due to the fact that most of the losses in water bodies in the period of 2000–2013 transitioned into wetlands in the entire subcontinent.

In the arid region, the largest part of the water bodies' losses was due to transitions of water bodies to wetlands, which represent about 30.5%, 36.5%, and 38.0% of the initial water bodies' extents in the 1975–2000, 2000–2013, and 1975–2013 periods, respectively (see Table A3 in the Appendix A). This was the case in some specific areas in Lac (Chad) and around Tombouctou and Mopti, located around the Inland Niger Delta in Mali, between 1975 and 2013 (see Figure 11), where a significant amount of water bodies transitioned to wetlands. A similar transition was also detected in the Senegal Basin.

The transitions of water bodies to other vegetation were 27.5%, 3.5%, and 25.5% in 1975–2000, 2000–2013, and 1975–2013, respectively (see Table 2). These were more observable around Lake Chad, particularly along the border between Lac (Chad) and Borno (Nigeria), as well as the border between Chad and Diffa in Niger (see Figure 11b1,b2).

### 4.4.6. Expansions of Artificial Water Bodies

Isolated gains in water bodies were detected mostly in the humid regions of our study area. We detected more water bodies' gains in the form of artificial dams and small reservoirs. Relative net gains of about 9.7% and 9.4% were observed in the humid region in 1975–2000 and 1975–2013, respectively, for water bodies (see Figure 5 and Table A2 in the Appendix A). The annual rate of change (0.4%) in water bodies in 1975–2000 (i.e., the drought era and soon after the drought) in the humid region was higher than the annual rate of change, i.e., 0% due to −0.3 relative net losses recorded in 2000–2013 (i.e., the post-drought era and recent period). About 0.2%, annual rate of change in water bodies was recorded in the humid region during the entire period (1975–2013). Nevertheless, these gains were not sufficient to compensate for the relative net losses in the entire West African subcontinent due to the massive relative net losses, i.e., −31.1%, −26.5%, and −57.6%, that we observed in the arid regions during 1975–2000, 2000–2013 and 1975–2013, respectively (see Figure 5 and Table A2 in the Appendix A). These losses in water bodies in the arid region were much larger than the losses,

i.e., −5.6%, −10.2%, and −15.7%, that we observed in the entire West African subcontinent during the same periods respectively (see Table A1 in the Appendix A).

The vast majority of the gains in water bodies observed in the humid region were at the expense of other vegetation (0.5% and 0.5%), forestland (0.9% and 0.9%), cropland (0.7% and 0.7%), and wetland (4.6% and 4.2%) in 1975–2000 and 1975–2013, respectively (see Table A3 in the Appendix A). Apart from the transition of other vegetation into settlements and croplands presented in the previous section, considerable fractions of other vegetation areas, i.e., 0.4%, 0.1%, and 0.3%, transitioned to water bodies in 1975–2000, 2000–2013, and 1975–2013 in the whole of West Africa, respectively (see Table 2). We identified countless sites where other vegetation areas were replaced by water bodies. This kind of LULC transition was more extensive in Ghana. A typical example was detected in Tain (Figure 10c1,c2) and Bamafele, as well as Kayes in Mali (see Figure 3), Shiroro in the Niger State of Nigeria, and many other areas (see Figure 9c1,c2).

In addition to the transitions of other vegetation to water bodies, we detected transitions of forestlands to water bodies. These transitions were also more pronounced in the humid regions of West Africa. Approximately, 0.9%, 0.1%, and 0.9% of the initial forestland areas transitioned into water bodies in 1975–2000, 2000–2013, and 1975–2013, respectively (see Table 2). Clear examples of these transitions were observed around Haut Sassandra and Nawa in Cote d'Ivoire (Figure 12a1,a2) as well as Shiroro in the Niger State of Nigeria (Figure 9c1,c2). As indicated earlier, the transitions of forestland to water bodies in the humid region were far larger between 1975 and 2000.

### 4.4.7. The Transition of Forestland to Other Vegetation

The transition of forestland to settlement, water bodies, cropland, other vegetation, and wetland were observed. The transitions into settlements, croplands, and water bodies constituted the greatest part of the forestland losses in our study area. These transitions have been described in the previous sections. However, one of the important forestland transitions, which may easily go unchecked but is vital in understanding forest degradation, is the transition of forestland to other vegetation. A fraction of forestland areas, i.e., 31.1%, 6.7%, and 30.4%, transitioned into other vegetation in 1975–2000, 2000–2013, and 1975–2013, respectively (see Table 2). The transition of forestland to other vegetation was higher during the period of 1975–2000 than 2000–2013 in the whole of West Africa (see Table 2) and the two sub-regions (humid and arid; see Table A3 in the Appendix A). This transition was particularly evident around Sassandra and Cavally in Cote D'Ivoire (Figure 12a1,a2), Kono and Kenema in Sierra Leone (Figure 12b1,b2), and Sedhiou in Senegal. In all cases, most of the forestland areas in 1975 were replaced by either cropland or other vegetation by 2013.

## 5. Discussion

### 5.1. Environmental Change in Africa

The findings from this study clearly indicate that LULC change under anthropic influence was one of the developments that triggered the environmental change in Africa after the severe drought of the 1970s and 1980s. We detected that the LULC transitions in West Africa before and after the period of 2000 occurred at the expense of natural vegetation (foreland and other vegetation) and water bodies due to settlement, cropland, wetland, and other LULC types' expansions.

Massive LULC change occurred mainly in the humid region in 1975–2000 (i.e., the drought era and soon after the drought) than the arid region. However, in 2000–2013 (i.e., the post-drought era and recent period), LULC change in the humid and the arid regions occurred at the same pace. This is because before the period of 2000, human activities in the arid region were relatively low, and crop farming, which had already gained firm ground in the humid region, largely depended on rainfall in the arid region [69]. According to UNEP [4], many pastoral farmers migrated to the humid region in search of greener pasture and fodder to feed their livestock during the severe drought era. This further increased the intensity of anthropic activities and expansions of human settlements in the humid region.

In contrast, the 2000–2013 period witnessed recovery of rainfall in the whole of West Africa [1–3] and a shift from rain-fed agriculture to irrigated agriculture in both the arid and the humid regions [69], thereby returning agriculture and other anthropic activities in both regions to normal. The annual rate of change for all the aggregated LULC types in our study was higher in the period between 2000 and 2013 (recent period) than the period between 1975 and 2000 (earlier period). The only exception was the annual rate of change in other LULC types, which was the reverse in the whole of West Africa and the arid region (see Table A1 in the Appendix A). This means that, in one way or another, the underlying drivers of LULC change were different in the arid and humid regions of West Africa at a given period.

Hickler et al., Seaquist et al., and Huber et al. [15–17] suggested that climate and natural variations such as variations in annual rainfall patterns and soil moisture are the major drivers of the observed vegetation dynamics and subsequent environmental changes in Africa. Eklundh and Olsson [8] found increasing rainfall during the period of 1982–1999 in the Sahel and linked it with vegetation recovery. This contradicts our findings. We found that the observed recovery of rainfall during this period did not lead to relative net gains in water bodies and natural vegetation in West Africa, especially in the arid (Sahel) region, during the three periods (1975–2000, 2000–2013, and 1975–2013). The relative net gain in water bodies we recorded in the humid region during the period of 1975–2000 is the only exception. However, our research further revealed that the expansion of water bodies in this region was due to massive expansion of artificial water bodies and did not lead to relative net gain in natural vegetation in this region (see Figure 5) and (Tables A1 and A2 in the Appendix A).

Overall vegetation loss was detected by our study at all the spatial scales of the analyses, which indicate deforestation and degradation of vegetation cover in West Africa. This contradicts the findings by Eklundh and Olsson [8] and Anyamba and Tucker [7], which reported overall re-greening (vegetation recovery) of the Sahel region of West Africa after the severe drought of the 1970s and 1980s. The observed vegetation recovery detected by Eklundh and Olsson [8] and Anyamba and Tucker [7] may be attributed to the massive cropland expansions (see Figures 5 and 6) discovered by our study. This is because the findings by Eklundh and Olsson [8] and Anyamba and Tucker [7] were based on trend analyses of NDVI as an indicator of vegetation conditions. NDVI is a good measure of green leafy display, but does not shed light on the underlying LULC change and its evolution. This has been noted by several authors (e.g., [9–12]) in some parts of West Africa. Similar conclusions were drawn by reviews on change detection methods by Coppin and Bauer, Lu et al., and Hussain et al. [48–50] and indicators of land degradation by Mbow et al. [44].

We found that many of the losses we detected in natural vegetation (forestland and other vegetation) during the three periods could not be fully and directly attributed to the severe drought or natural factors alone. Natural vegetation losses, particularly in the humid region, were due to conversion to cropland, settlements, other LULC types and artificial water bodies. Such transitions may be linked with anthropic activities and were more pronounced in the southern humid regions, e.g., Cote D'Ivoire, Ghana, and Nigeria, with high population growth. Population growth exacerbates the transition of natural vegetation to settlement and cropland [14]. This suggests that variation in annual rainfall pattern is not the major driver of vegetation dynamics in the entire West African subcontinent and the two sub-regions (humid and arid), though this assertion may hold for some local areas. These findings are in agreement with the NDVI trend analyses, supplemented with LULC change analyses by Boschetti et al. [10] and Leroux et al. [9], which highlighted that vegetation change in Africa cannot be explained by a variation in rainfall pattern alone and suggested that anthropic activities are also responsible for the observed changes in vegetation. Similar findings were observed by Helldén and Tottrup [83].

Comparatively, the findings from our study are also consistent with the findings of Boschetti et al. [10], Leroux et al. [9], and Rasmussen et al. [84], who reported degradation of vegetation cover in some parts of West Africa and attributed the isolated increase in vegetation cover to expansion/intensification of agricultural activities. The meta-analysis by Vliet et al. [85] also highlighted the massive expansions of cropland in some parts of West Africa. Nevertheless, our findings contradict

some parts of the findings of Vliet et al. [85], who stated that, despite population growth, crop area has been stable in areas where arable lands are limited, and farmers have adopted new technology to intensify their farmland.

Our study revealed widespread cropland expansion over the entire West African subcontinent during all the three different time periods of our analyses, except that the cropland expansion in the densely populated humid regions was higher than in the arid regions with lower population density and unfavorable climate conditions [69]. The accelerating rate of cropland expansions stems from the fact that farming is the major occupation in West Africa and a major contributor of the gross domestic product (GDP) of this continent [69]. In view of this, farmers are often given financial incentives and government subsidies in the form of pesticides, fertilizers, and seedlings to boost crop production for export and subsistence. Field studies in some parts of Ghana (e.g., [25,33]) and Cote D'Ivoire (e.g., [86]) have documented a shift from subsistence farming to large scale cash crop (e.g., cocoa, oil palm, rubber, and coffee) farming. These findings are consistent with the findings from our study, as we observed massive expansions of plantations in the form of cash crops in the study area during the period of 1975–2013 on the original USGS maps with 24 classes in West Africa (see Figures 2 and A1 in the Appendix A).

The large scale expansions of cropland may lead to reductions in land areas available for farming, competition among farmers, and land conflicts. One important question that needs to be addressed is whether the expansion of cropland detected by our research resulted in higher crop yields. Lower crop yields at the expense of forest and other natural resources over a large area may have a long term negative impact on the future food security and livelihood of the local communities who depend on forest resources for their survival. The way forward is implementation of sustainable agricultural practices such as effective soil management, e.g., application of cow dung and organic manure to restore soil fertility; cultivation of climate-smart and drought-resistant crops; sensitizing farmers to good farming practices; and alternative sources of livelihood for the farmers [25,28–35]. Further research to assess crop productivity and yield is also recommended in this region.

Moreover, apart from the exportation of cash crops by the West African countries, other rich natural resources in the form of minerals (e.g., gold, bauxite, diamond, and so forth) and timber are widely exported from the subcontinent to countries outside Africa. Nonetheless, African countries are not well-equipped to participate in the global market, and thus exporting of such minerals and timber may have serious implications for the environment [87–91]. One of the ways to tackle this problem in Africa is to strengthen the agriculture value chain and adoption of natural resource regeneration and bio-energy technologies (which will ease the pressure on hydro-power generation from natural water bodies) to ensure maximum benefit of the natural resources under minimal environmental damage [92].

On the other hand, even though settlement expansions doubled in the study area during the period of this analysis, we found that the proportion of the total land area covered by settlement in the final year (2013) was significantly low (0.7%). This signifies that the expansion of settlement with respect to the total land area in Africa has not reached its equilibrium, and hence its contribution to the environmental change on the entire continent is localized. The possible reason for the relatively low fractional abundance of settlement in West Africa could be the extended family system in Africa, which permits a considerable number of family members to live in the same compound house. This kind of family system increases the total number of inhabitants per unit area and may lead to urbanization and subsequent urban sprawl in the subcontinent. Urbanization and subsequent urban sprawl may have serious implications for the environment. Another factor that must be taken into consideration in West Africa is that urbanization and urban sprawl are not evenly distributed. They are more or less centralized in some few mega-cities. Nevertheless, the current trend we observed suggests that settlements will continue to increase in the future due to the high fertility rate of women in Africa, changes in lifestyle of the affluent population, increasing demand on infrastructure, and competition

for space for emergent industrial development. All these developments may put additional pressure on the West African landscape and may exacerbate environmental change in the future [93,94].

*5.2. LULC Transitions and Underlying Driving Factors*

The LULC transitions we observed in our study area were in agreement with the findings of Lambin et al. [45], who stated that LULC transitions in each region depend on specific conditions such as socio-economic factors, e.g., population density, poverty rate, human lifestyle change, climatic conditions, and specific policies in each location at a given time. These specific conditions also influenced the direction and the type of the LULC transitions we observed in different locations of our study area at a given time. Lambin et al. [45] broadly grouped the major underlying drivers of LULC transitions in West Africa into anthropic and climate drivers. Essentially, most of the LULC transitions we detected in West Africa during the 38 years period, i.e., 1975–2013, may be considered as human-induced LULC transitions, which may be linked with underlying anthropic drivers. In some instances, we detected some LULC transitions that may be linked with climate drivers or a combination of the two drivers.

Anthropic Drivers

The massive expansions of settlements and cropland at the expense of natural vegetation were larger in the densely populated and humid countries such as Nigeria, Ghana, Cote D' Ivoire, Togo, and Benin. These densely populated humid countries have favorable climate conditions characterized by rich and dense forest, which often forms a canopy and understory, coupled with easy access to road infrastructure, markets, and urban services, unlike the arid areas located in the northern part of our study area with unfavorable climate conditions and scattered vegetation, where our study revealed fragmented LULC changes [69].

Several previous studies [25,28–35] in some local hotspots of our study area have attributed the massive expansions of settlements and cropland to population growth, high demand for accommodation, and the quest to ensure food security. Gray, Leblanc et al., and Paré et al. [22,95,96] pointed out that simultaneous expansions of settlements and cropland in different locations in West Africa may confirm that cultivated areas in West Africa expand in parallel with population growth. This suggests human influence on the transition process. Other anthropic drivers, such as proximity of natural vegetation to urban settlements, in-country migration and over-harvesting of forest resources such as timber, fruits, and herbs for medicines, have been cited as factors that played an instrumental role in the natural vegetation losses in these regions [25,32–34,97–99]. Some aspects of land governance, such as land tenure insecurity and weak policy dialogue among stakeholders, have also been cited as some of the anthropic drivers of natural vegetation losses in the study area [25,28–35].

A number of the LULC transitions we detected in the study area have also been linked with implementation of some specific policies and initiatives under anthropic influence, such as "National Government Policies", the "Taungya Policy", the "REDD+ Policy", and the "Farmer-Managed Natural Resource Regeneration Initiative", adopted in the study area. For instance, part of the settlement expansions we detected in some cities of Nigeria (see Figure 8) have been linked with national policy changes, such as the establishment of the new Federal capital and the movement of the government offices to Abuja. This relocation saw many farming communities and areas earmarked for forest protection and agriculture being converted to settlements due to an influx of local migrants in the capital city. Although the future prospects of urbanization such as creation of market opportunities for farm produce, alternative sources of employment, and income generation are promising, urban sprawl may disrupt agricultural and forest economies in West Africa [28,29,34,99,100]. Some studies [25,98] have proposed compact city structures, such as vertical building systems and strict implementation of urban planning laws, to curb the massive expansion of settlements in West Africa.

The "Taungya Policy", which was introduced in some forest zones, e.g., Ghana, Cote-D'Ivoire, and Nigeria, following the drought of the 1970s and 1980s, explains part of the observed expansion of settlements and cropland at the expense of natural vegetation. The goal of the policy was to rehabilitate degraded forestland by introducing agro-forestry practices to benefit the local farmers. Literature suggests that the policy scheme lost focus by attracting local migrant farmers and resulted in increased pressure on vegetation cover and other natural resources in the protected forest areas [69,99,101–103].

As a result, many forest zones in some parts of West Africa are currently under protection and human activities are strictly regulated due to the recent piloting of the "REDD+'s Policy". Though we observed a massive forestland loss at all the spatiotemporal scales of our study, the "REDD+'s Policy" [36–38] may explain the differences we observed in the transition of forestland to other vegetation during the earlier 1975–2000 and recent 2000–2013 periods (see Table 2 and Table A3 in the Appendix A). The transition of forestland to other vegetation is an indication of forest degradation. This transition was larger during the period of 1975–2000 than the period of 2000–2013 in the whole of West Africa (see Table 2) and the two sub-regions (humid and arid; see Table A3 in the Appendix A). This may suggest that the REDD+'s pilot projects have been effective to some extent.

In contrast, we found large scale transition of forestland and other vegetation to "other LULC types" during the period of 2000–2013 around the same locations in the humid region, where we observed the reductions of forestland's transition to other vegetation. The possible explanation could be a shift from illegal and selective logging of trees in the forest to illegal mining of gold, bauxite, and other minerals often referred as "galamsey" [104], as open-mine fields were one of the sub-categories of "other LULC types" in our study. "Galamsey" has the potential to directly degrade the natural vegetation and may lead to gully erosion and leaching of nutrients with subsequent decline in soil fertility and biodiversity. In addition, our research observed massive expansions of settlement and cropland at the expense of natural vegetation around the same locations where the massive expansions of the "other LULC types" were detected. These developments indicate severe deforestation and land degradation with negative implications on future carbon stock and climate change mitigation [36–38].

Acheampong et al. [33], Kleemann et al. [98], Obodai et al. [104], and Koranteng et al. [105] have also documented deforestation as a result of agricultural (cropland) and settlement expansions as well as illegal mining in some parts of West Africa where we detected massive deforestation. This calls for enactment of strict rules and policies as well as sustainable land use planning regulations to maintain a balance between food security, socio-economic development, and natural resource management in West Africa. It is expected that full implementation of the "REDD+'s Policy" will lead to reductions in deforestation and degradation, sustainable regeneration of trees, carbon stock enhancement, and financial incentives for land users in the future [38].

However, the prevailing conditions and the anthropic developments in West Africa suggest that the vast majority of the people depend on forest resources for their survival. According to Cacho et al. [106], the alternative land uses for forestland such as exploitation for food, cash crops, medicine, illegal mining for gold, and timber may be more profitable than the incentives to avoid deforestation offered by the REDD+'s policy makers to land users. Conservation of forest means depriving these indigenous people and land users their sources of livelihood [97,107]. In order to reduce the adverse effects of REDD+ projects and for effective and sustainable implementation of the project, the process of co-benefits' and leakages', especially cropland's impact on the forestland areas earmarked for the project has to be analyzed and understood. Additionally, the profitability gap of maintaining cropland (e.g., cash crops) and compliance with REDD+ intervention has to be analyzed [108–110]. Such analyses, according to Pirard [108] and Golub [110] will ensure correct estimation of carbon financial credit to land users for avoiding deforestation, a low-cost emission strategy, and active community and stakeholder involvement.

Some studies (e.g., [35,56,111]) have documented the achievements of "Farmer-Managed Initiatives and Resilience Building Strategies" adopted by communities and individuals to recover degraded landscapes for farming in West Africa. Successful implementation of such resilience building in the

Sahel (arid region) context is the "Farmer-Led Natural Regeneration of Trees" in the Maradi and Zinder regions of Niger, Central Plateau in Burkina Faso, and Seno Plain in Mali. According to Tappan and Mcgahuey, Reij et al., and Sendzimir et al. [35,56,111], these interventions transformed degraded landscapes into fertile land for farming, which in turn increased drought resistance, enhanced crop productivity, improved food security, and alleviated poverty in the mid-1980s. Our study also detected and confirmed cropland expansions in the aforementioned regions in West Africa (see Figure 8a1,a2), and Figure 9a1,a2,b1,b2). Sendzimir et al. [35] highlighted that the introduction of understory forest in the Zinder and Maradi regions of Niger helped to restore the rainfall pattern and soil fertility, leading to improved performance of crops and pasture. This confirms that sustainable management of forest, good farming practices, and community involvement can help communities to recover from severe land degradation.

The anthropic activities in the drought era also influenced the relative net gain in water bodies we observed in the humid region of our study area. Artificial dams and small reservoirs were observed at the fringes of some of the areas where we detected natural vegetation losses. The transition of natural vegetation to water bodies may indicate natural flooding or anthropogenic development of pounds and small reservoirs for cropland irrigation, dams for hydro-power generation, urban supply and consumption, as well as other economic purposes [23,24,27,56,86,95,107,112–118]. These developments could possibly be explained as adaptation measures adopted in some parts of West Africa to mitigate the impact of the severe drought of the 1970s and the 1980s. Previous studies by Obour et al. [115] and Hausermann [117] confirm that the expansion of water bodies we detected in the Tain District (see Figure 10c1,c2) of Ghana is due to a dam (Bui dam) constructed on Ghana's Black Volta for hydro-power generation. Hausermann [117] noted that, though the dam was not originally intended for irrigation purposes, it later attracted local migrant farmers from the surrounding cities and part of the economic benefits of the dam shifted to fishing and cropland irrigation.

Additionally, a detailed literature review confirmed that one of the new water bodies we identified around Sassandra in Cote D'Ivoire (see Figure 12a1,a2) is "Buyo Dam", built across the Sassandra river in the 1980s to promote hydropower generation, intensive agriculture for subsistence, and cash crops for export [86]. According to Conway [86], this attracted local migrant farmers, leading to a population increase, which further intensified agriculture expansion in this region. Apart from the unprecedented agricultural expansion that threatens the environment, unregulated use of pesticides and overuse of fertilizers and poor water quality due to waste disposal into the dam have also been reported [107]. All these developments came at the expense of forestland. The year of the dam's construction coincided with the period of the data analyses of our research [86]. Therefore, strategies for sustainable water management are urgently needed in this region.

Climate Drivers

According to Niang et al. [21], irregular rainfall caused the widespread vegetation stress and subsequent drying out of vegetation during the severe drought of the 1980s in the arid (Sahel) region of West Africa. This may explain the higher annual rate of transition of other vegetation to other LULC types we detected in the arid region during 1975–2000 as compared to the lower annual rate in the same region during 2000–2013. Niang et al. [21] documented the disappearance of vegetation cover after the 1990s in Mauritania around the same area where we detected the replacement of other vegetation by other LULC types. The field study by Niang et al. [21] complemented with RS surveys documented an increase in "sand dunes" and a decline in forestland and other vegetation due to the persistent drought. The unprecedented expansion of other LULC types is one of the key alert signals of land degradation and may have negative impacts on the environment. Nutrients in the soil are depleted when the vegetation cover is wiped away and the soil is exposed. Additionally, a transition like this may threaten biodiversity with subsequent extinction of flora and fauna. According to Nicholson, Tucker et al., Hulme and UNEP [1–4], similar events triggered land degradation and desertification in Africa, particularly in the arid (Sahel) region, in the 1970s and 1980s.

Combination of Climate and Anthropic Drivers

Combinations of climate and anthropic drivers also played a significant role in the LULC transitions we observed. Reij et al. [111] and CILSS [69] linked the cropland loss we detected in the peanut basins of Senegal (e.g., Diourbel, Louga) with anthropic drivers, such as abandonment of farmlands due to declining prices in peanuts, subsequent fallowing, and natural regeneration of trees. The former forced some farmers to migrate to the nearby cities such as Touba and Dakar. Similar findings were detected along the border between Senegal and Guinea-Bissau by Costa and Cabral [26], who linked the decline in agricultural practices with anthropic drivers such as the exodus of the rural people during the Casamance struggles for independence. The same authors argued that different land management policies led to different landscape evolution in Senegal and Guinea Bissau.

Additionally, a combination of anthropic and climate drivers may explain the shrinking of water bodies detected in the arid regions of our study area during the 1970s and 2000s, while massive gains in water bodies were detected in the humid regions during the same period. These differences may be attributed to climate variation across different ecological zones and differences in the intensity of human activities. CILSS [69] attributed the shrinking of water bodies we detected in Lake Chad Basin to the combined effect of decreasing rainfall and the repeated episodes of the severe drought, as well as water withdrawal for irrigation.

Coe and Foley [27] also confirmed that Lake Chad has lost about 90% of its initial area over 40 years. Coe and Foley [27] attributed about 50% of the observed decrease in the Lake Chad area as early as the 1960s and 1970s to anthropic drivers such as population growth, migration, deforestation, over-fishing, uncoordinated construction of dams, and human water uses such as water withdrawal for irrigation and consumption. Coe and Foley [27] further linked the disappearance of Lake Chad to irregular rainfall; the recurrent drought; the uniqueness of the Lake's bathymetry, which permits it to break into smaller lakes; and the inability of the lake to fully recover after the severe drought of the 1970s and 1980s. Conversely, our results clearly show that withdrawal for irrigation is the key driver, as irrigated agriculture expansions were noticed at the fringes of the Lake Chad. A field survey by Oyebande et al. [57] at the Nigeria portion of the Lake also attributed about 45% of the loss in water bodies to anthropic factors and decreasing rainfall. Niel et al. [114] documented cases of decreasing annual precipitation in the area of the Lake that stretches to the south. According to Niel et al. [114], the deeper portion of the Lake, which extends northwards, dried completely in the 1980s.

Furthermore, a field study by Tappan and Mcgahuey [56] confirmed the expansion and diversification of agriculture in Bani, a city located downstream of Mopti and Segou in the Niger Delta in Mali (see Figure 11). In this area, we identified hotspots of wetland transitions into cropland. Anthropic activities such as irrigated rice cropping, rain-fed agriculture, grazing and browsing by livestock, as well fuel wood extraction at the expense of wetlands and water bodies in the same region have been documented by Lutz et al. [24]. The effects of population growth, urbanization, and agriculture expansion on ground water extraction in the same area in Mali have been estimated by Lutz et al. [24]. According to Lutz et al. [24], these activities led to higher rates of ground water extraction.

The relative net gains in wetlands we observed in the whole of West Africa have also been linked with anthropic drivers such as the rehabilitation projects in the form of re-flooding of dried-up floodplains. These interventions, according to Scholte et al. [119], Loth [120], and Hamerlynck et al. [121], were initiated in the 1990s, with the expansion of wetlands at the expense of water bodies being due to a combination of climate and anthropic drivers. Expansion of wetlands is good for the aquatic habitat and health, but the expansion at the expense of water bodies and natural vegetation must be duly brought under control.

Some authors contend that the current shift from rain-fed to irrigated agriculture in some parts of Africa may lead to widespread agricultural drought. It is expected that the renewed attention to the "Great Green Wall" may help to avert the devastating impact of climate and anthropic activities on desertification, soil degradation, and greenhouse gas emission [41]. The "Great Green Wall" was initially advocated as a means of reducing desertification in the Sahel through the planting of a broad continuous band of trees from Senegal to Djibouti [40]. It is likely that the Sahel "Great Green Wall

initiative" will reap some benefits due to the influence of vegetation on the terrestrial water cycle. Examples of such influence are the projected increase in the number of rainy days by +9% and the potential of the initiative to mitigate the intensity of heavy rain events over the Sahel while decreasing the extreme drought spells [40].

*5.3. Global and Continental EO LULC Data*

LULC change detection results depend on the accuracy of the LULC data [50], but prior to our study, the consistency of the USGS LULC data [69] with the other global and continental LULC data (see Section 1.3 of the Introduction) had not been assessed. To fill this gap, we assessed the accuracy and the reliability of the USGS datasets by comparing the spatial distribution and the LULC statistics of the 2000 LULC map with some of the existing global and continental LULC data, i.e., GLC-30m, MODIS-MCD12Q1, and ESA-CCI-LC. The LULC map in the year 2000 was chosen for this comparison because of the absence of cloud cover on the map unlike the other LULC maps for different years. Additionally, all the remaining LULC data, except MODIS, had LULC maps in the year 2000. In the case of MODIS, the LULC data in the year 2001 were used for this comparison (unpublished research results by the authors of this research).

We found large discrepancies between the different LULC data. To evaluate the LULC data, previous studies have used estimates based on in-situ observations from crowd-sourcing portals such as the Geo-Wiki portal, Google Earth Engine [63,64] and FAO [73] FAOSTAT (accessible at http://www.fao.org/faostat/en/#data/RL). We estimated the root mean squared error (RMSE) of cropland and forestland area from each dataset against the national cropland and forestland statistics provided by the FAO [73] (http://www.fao.org/faostat/en/#data/RL) for the year 2000. FAOSTAT was used as a benchmark because the USGS data to our knowledge had already been validated against crowd-sourcing datasets [69].

The forestland and cropland estimates from the USGS LULC map in the year 2000 were closer to the FAOSTATS than the other global LULC maps. The relative RMSE of cropland estimates were 32.6%, 46.6%, 60.84%, and 184.86% for USGS, GLC-30m, MODIS-MCD12Q1, and ESA, respectively. The relative RMSE of forestland estimates were 58.74%, 67.8%, 88.03%, and 70.97% for USGS, GLC-30m, MODIS-MCD12Q1, and ESA-CCI-LC, respectively. When the national forestland and cropland estimates of the FAOSTATS were merged into a single reference data, relative RMSE estimates were 45.7%, 57.19%, 74.4%, and 112.41% for USGS, GLC-30m, MODIS-MCD12Q1, and ESA-CCI-LC respectively. The comparison demonstrated that the USGS LULC data agreed better with the national level FAOSTATS in all instances than all the other Global LULC data, and hence our decision to use the USGS LULC data for this study was justified. It remains to be clarified why all the LULC datasets gave either overestimates or underestimates of the forestland and cropland areas.

The discrepancies we detected among the various LULC data may be linked with class definitions and classification schemes adopted for the development of the different LULC maps. We developed a common classification scheme to compare all the LULC data, but this did not remove possible sources of uncertainties. Examples of the latter are shadows being mapped as water bodies, and agricultural areas being underrepresented and mapped as natural vegetation in some of the datasets. Comparisons of some continental and global LULC data by Tsendbazar et al. [66] also revealed that remotely-sensed global land cover products from different sources may exhibit large discrepancies in spatial distribution and high uncertainty in the estimated area. Vliet et al. [85] also detected discrepancies in cropland estimates from scientific literature on cropland estimates in Africa.

FAOSTATS is not without limitations, e.g., lack of completeness of the national statistics has been documented (http://www.fao.org/faostat/en/#data/RL). This may explain the large relative RMSE we obtained in some cases for the LULC data. Accurate LULC information is critical in LULC change detection and global environmental studies [122]). USGS attempted to triangulate and validate the LULC data we used for the analyses with thousands of aerial photographs and high resolution satellite images, field surveys, and subsequent independent reviews of the data by the respective country team

of image interpreters [78]. This explains the better agreement with FAOSTATS as compared to the other LULC data.

*5.4. The Study Approach*

We aggregated and reclassified the 24 LULC classes in the original datasets generated by CILSS [69] into seven LULC classes, i.e., cropland, forestland, other vegetation, wetland, water bodies, settlements, and other LULC types, before the detailed transitions analyses. The new classification scheme was used as a common legend for all the existing global LULC data compared in the unpublished research work by the authors of this study. Without the class aggregation and reclassification, the area of changes and percentage (%) of the transitions suffered by the 24 LULC classes would have been very small in number, much smaller than the error of estimates, thus making the evaluation of the transitions negligible.

Our study applied the post-classification change detection technique, because the datasets we used for the transitions analyses had already been classified by CILSS [69]. The post-classification change detection technique has the advantages of minimizing the problem of radiometric calibration and other atmospheric influences between different dates. The technique was effective in generating a complete change matrix for identifying the nature of the LULC changes, thereby differentiating cropland changes from natural vegetation changes. Such a differentiation was not possible using the frequently used change detection methods in Africa such as image differencing for NDVI trend analysis [48–50].

The major limitation of the post-classification change detection technique is that it requires large training samples and expert knowledge. Additionally, the final accuracy of the change output depends on the accuracy of the initial classification of the individual images [48–50]. These limitations were addressed by CILSS [69] during the LULC data development process. According to Tappan et al. [78], experts of image interpreters in the respective countries in West Africa improved the accuracy of the datasets by validating them against multiple source LULC data and field samples. Data integration and image fusion are able to improve the accuracy of satellite EO and RS data [123].

Despite that, some level of uncertainty remained. The use of the satellite RS technique to capture land surface information for LULC change detection is limited in tropical regions due to dense forest canopy and intermingling crowns, cloud cover, and saturation of RS signals [38]. As a result, our study found missing data values and cloud pixels on the LULC data. These were classified as a separate LULC class on the original USGS LULC maps (see Figure A1 in the Appendix A) by CILSS [69]. The greater proportion of these pixels covered some parts of the original LULC map for 1975 known to be forestland and settlements based on expert knowledge in the humid region (e.g., Ghana). During the period of 2000, the vast majority of the areas initially covered by missing data and cloud pixels in 1975 were replaced by cropland. To reduce this uncertainty, these pixels were masked from all the three LULC maps by our study. This process either underestimated or overestimated the extent of the LULC changes in the affected LULC classes.

Notwithstanding, cropland and forestland estimates from our study were in line with the estimates form the sample-based LULC mapping approach by Brink and Eva [59]. Brink and Eva [59] detected expansion of agricultural land in 1975 and 2000 in Sub-Saharan Africa. They attributed this expansion to the fact that agricultural activities returned to normal growth after the drought of the 1970s. Although the study by Brink and Eva [59] covered the whole of Africa, most of the LULC changes we detected during 1975–2000 were in agreement with their findings. Between 1975 and 2000, we detected a relative net gain of 56.8% at an annual rate of change of 2.3% in cropland and relative net loss of −17.2% at an annual rate of change of −0.7% in forestland (see Table A1 in the Appendix A) respectively. The estimates by Brink and Eva [59] during the same periods were also a relative net gain of 57.3% at an annual rate of change of 2.3% for agriculture land and a relative net loss of −16.3% at an annual rate of change of −0.7% for forestland. This is partly due to the fact that most of the areas in Africa that were unmapped by our study exhibit stable vegetation cover changes over time [78].

On the other hand, the LULC change estimates with the sample-based approach by Vittek et al. [58] cannot be directly compared with our findings due to the differences in the temporal coverage of the LULC maps and the discrepancies between the LULC classification systems adopted by the two studies. Vittek et al. [58] used 1975 and 1990 LULC data for their analyses, while we used 1975, 2000, and 2013 LULC data for our analyses. Additionally, the LULC class "other vegetation" defined by Vittek et al. [58] included other non-forest vegetation, agricultural land, and bare land, while our research categorized each of these LULC types into distinctive classes. Nevertheless, the general trends in the transitions detected by the two studies were similar. Forestland losses were detected around Nigeria, Ghana, Cote D'Ivoire, and Liberia by our current study. Vittek et al. [58] also detected forestland losses around the same areas in West Africa.

Furthermore, one of the main objectives of this study was to examine the underlying driving factors of LULC transitions and environmental change in West Africa and two sub-regions (humid and arid). Such analysis is often done by developing a relationship (regression model) between the observed LULC and a set of socio-economic and climate indicators [124]. The current study focused on analyzing the relative impacts of the various drivers by linking the observed LULC transitions at some hotspots with the underlying drivers documented by previous literature. Therefore, further studies will be required to test the statistical significance of these drivers. That is the next step of this study.

## 6. Conclusions

We analyzed the extent, magnitude, and nature of LULC transitions as well as their major drivers after the severe drought of the 1970s and 1980s in West Africa with EO and GIS applications. The most consistent LULC transitions we observed were natural vegetation encroachment by cropland and human settlements and in some case other LULC types. The major conclusion that can be drawn from this study is that deforestation and degradation of natural vegetation cover are widespread in the whole of West Africa, and they are being driven by human settlements and cropland encroachment. This phenomenon can be classified as human-induced LULC change. Human settlements were found to expand in parallel with cropland, which represents increased human demand for food to support the increasing population. Deforestation and degradation of the natural vegetation cover may have serious implications on the future carbon stock enhancement and emission reduction of the subcontinent and the world at large. Although an overall vegetation cover loss was documented, some specific areas experienced gains in vegetation cover. There were some areas where cropland losses were also noticed.

The intensity of the LULC changes we detected in the period of 2000–2013 (i.e., the post-drought era and recent period) was higher than the LULC changes we detected in 1975–2000 (i.e., the drought era and soon after). This indicates that additional factors apart from seasonal variations drive LULC changes in West Africa. Furthermore, the LULC changes we detected were more pronounced in the humid region (Sudanian, Guinean, and Guineo-Congolian) than the arid (Sahel) region. The transitions we detected in each region also varied through time and space. These differences in the LULC transitions we detected in the two sub-regions (humid and arid) as well as at some specific locations suggest that no single trend applies to the whole of West Africa, i.e., LULC transitions in West Africa are location specific. The vast majority of the LULC transitions we detected were due to underlying anthropic drivers. There were some instances where we detected LULC transitions in response to a combination of climate and anthropic drivers. The findings from this study indicate how local processes driven by anthropic activities lead to changes at the regional, national, and consequently at the continental level.

The proposed replanting of a continuous band of trees in the arid regions and the re-afforestation in the humid regions may help to address deforestation and degradation of natural vegetation in West Africa. We conclude that efficient implementation of national and international land use policies in West Africa requires active community involvement, stakeholder engagement, and appropriate laws and regulations to restrict the conversions and maintain a balance between food security, socio-economic development, and sustainable natural resource management. The shrinking of water bodies we

observed in the study area may also have serious implications for water balance. This calls for strict water management policies to protect the water bodies in West Africa. Land use policies such as the "REDD + Policy" and the "Great Green Wall Initiative" in West Africa must take into consideration the opportunity costs and profitability gaps of the alternative land uses to ensure co-benefits, compliance, and sustainability, while avoiding leakages.

The study demonstrated the potential to extract useful earth surface information from long term EO LULC data with current innovations in GIS applications for LULC transitions analyses at the continental scale. Our approach disentangled the LULC impacts of climate and anthropic drivers. The findings will be useful for natural resources managers, land planners, government institutions, policy makers, and other stakeholders for targeting and allocating resources and other land use interventions such as the "REDD+ Policy" and the "Great Green Wall Initiative in West Africa". Future research in this region must be focused on employing different metrics, e.g., night-time light from EO image data, to confirm anthropic activities intensification in parallel with the expansion of settlements observed by this study. Statistical significance testing of the major drivers of the observed LULC change is also recommended. Studies to relate this LULC change analysis to time series of rainfall and population density data are further required to validate the observed human-induced LULC change. Comparison of the LULC change results with NDVI analyses as well as the impact of this LULC change on biodiversity and water balance must also be given careful consideration.

**Author Contributions:** Conceptualization: B.A.B., L.J., and M.M.; data curation: B.A.B.; formal analysis: B.A.B., L.J., and M.M.; funding acquisition: L.J. and M.M.; investigation: B.A.B., L.J., and M.M.; methodology: B.A.B., L.J., and M.M.; project administration: L.J. and M.M.; resources: L.J. and M.M.; software: B.A.B.; supervision: L.J. and M.M.; validation: B.A.B.; visualization: B.A.B.; writing—original draft: B.A.B.; writing—review and editing: B.A.B., L.J., M.M., J.Z., and Y.Z. All authors have read and agreed to the published version of the manuscript.

**Funding:** This project was jointly funded by the National Natural Science Foundation of China (NSFC) & United Nations Environmental Program (UNEP) (Grant No. 41661144022), the Chinese Academy of Sciences President's International Fellowship Initiative (Grant No. 2020VTA0001), the MOST High Level Foreign Expert Program (Grant No. G20190161018) and the Chinese Government Scholarship Council (CSC) (Grant No. 2018SLJ023247).

**Acknowledgments:** B.A.B. is grateful to CILSS–USGS for making available the datasets analyzed in this study. All the authors wish to express their appreciation for the support provided in many ways by the staff of AIR, particularly, X.R. Liu, X. Li, and M. Jiang, and other research fellows, particularly X.T. Yuan, P. Pani, and A. Bennour. B.A.B. is grateful to James Asenso Barnieh and I.K.M. Quaye for encouragement and support.

**Conflicts of Interest:** The authors declare no conflict of interest. The funders had no role in the design of the study; in the collection, analyses, or interpretation of data; in the writing of the manuscript; or in the decision to publish the results.

## Appendix A

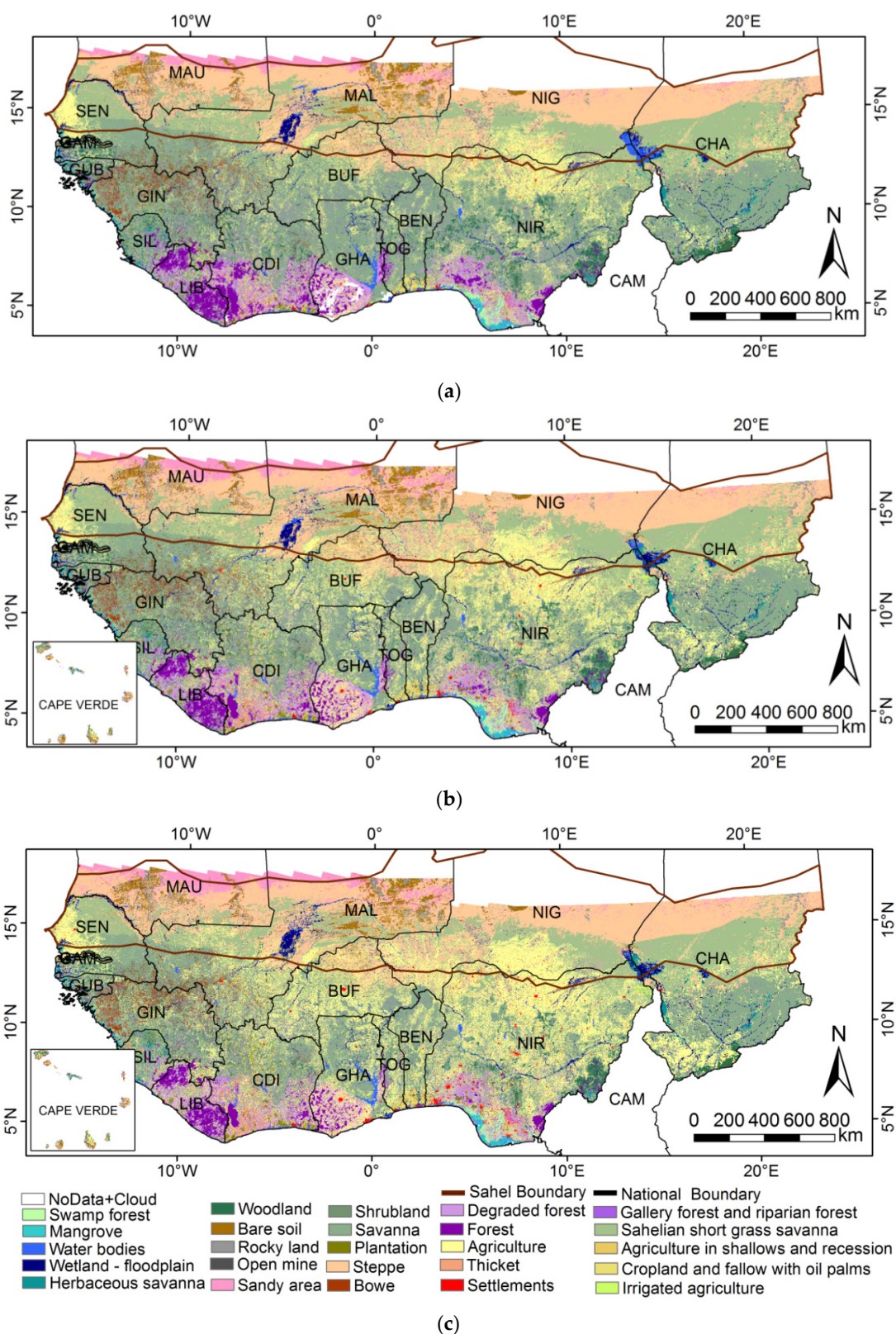

(**a**)

(**b**)

(**c**)

**Figure A1.** The USGS original Land Use Land Cover (LULC) maps of West Africa in: (**a**) 1975; (**b**) 2000; and (**c**) 2013.

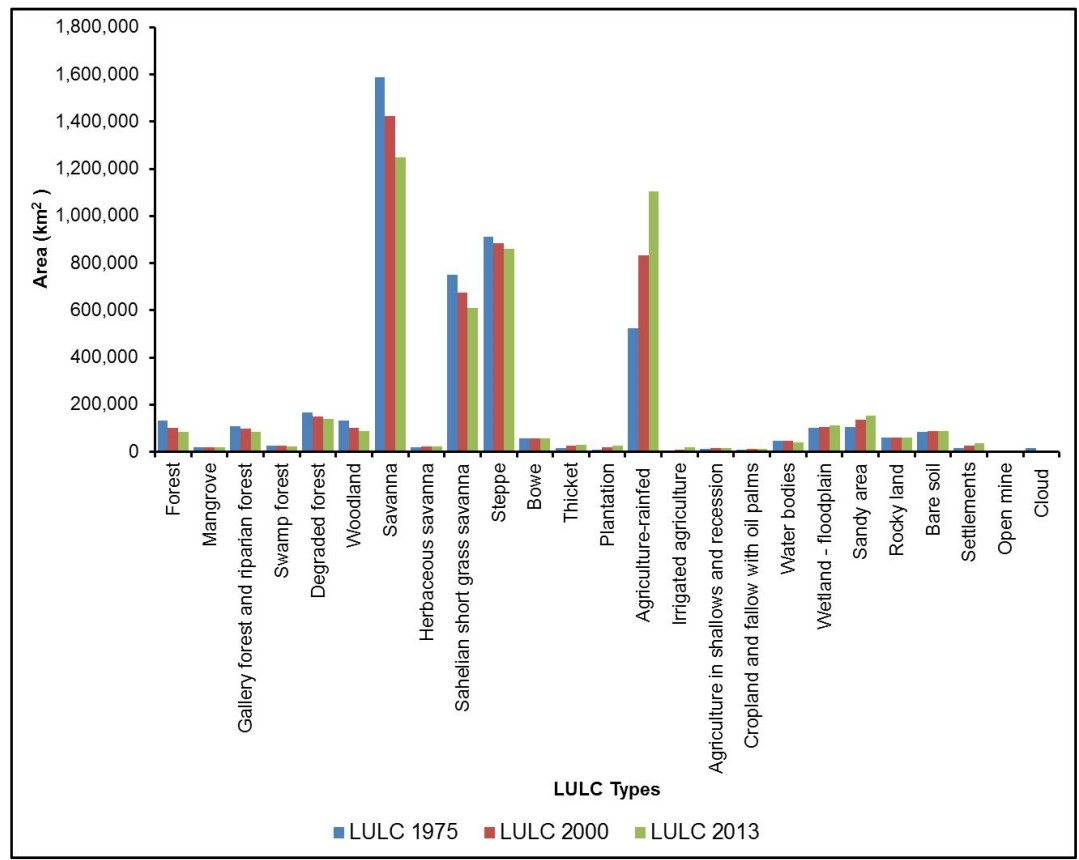

**Figure A2.** The area extent of the USGS original 24 Land Use Land Cover (LULC) classes/types (1975, 2000 and 2013) in West Africa.

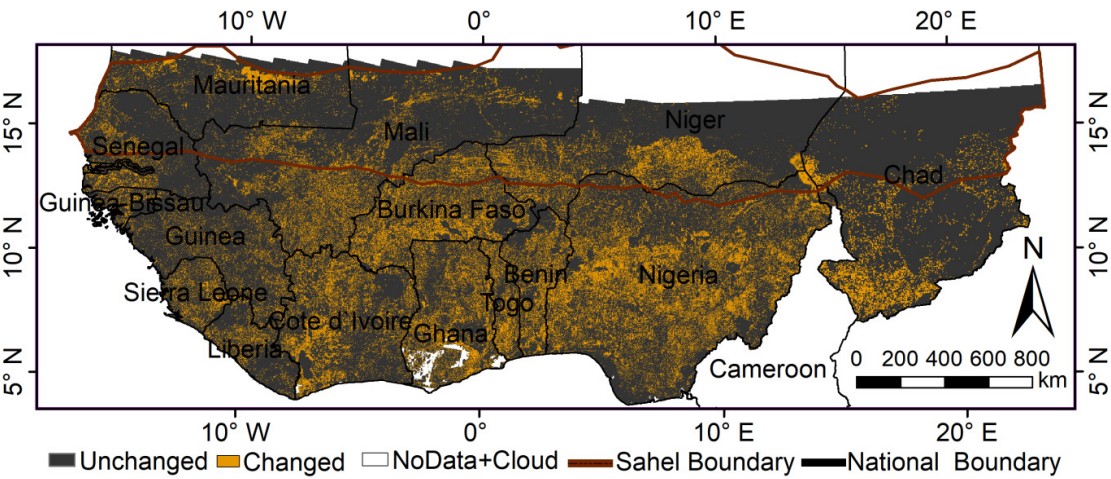

**Figure A3.** Land Use Land Cover (LULC) change map of West Africa between 1975 and 2013.

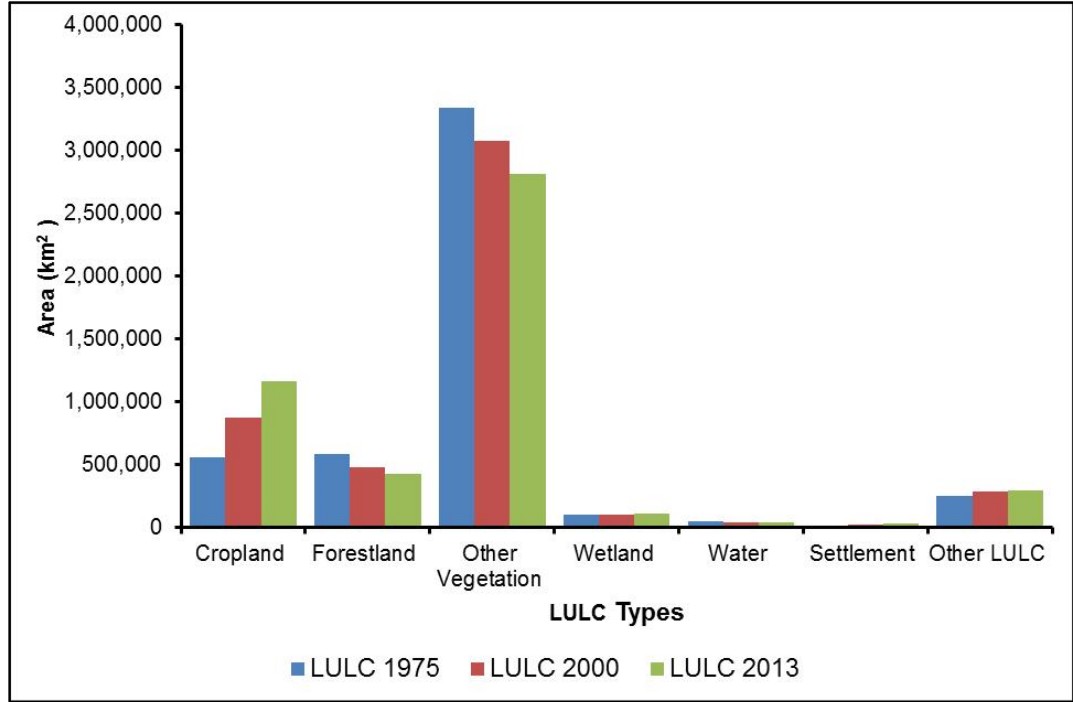

**Figure A4.** The area extent of the seven aggregated Land Use Land Cover (LULC) classes/types in West Africa at three time intervals (1975, 2000, and 2013).

**Table A1.** The net change in area/extent (km²) and the net relative loss/gain (%) of the seven aggregated Land Use Land Cover (LULC) classes/types in West Africa (1975–2000, 2000–2013, and 1975–2013). Each interval/period of the LULC transitions is highlighted in "grey".

| LULC Types | Area (km²) | Area (km²) | Net Loss/Gain (km²) | Net Loss/Gain (%) | Annual Rate of Change (%) |
|---|---|---|---|---|---|
| **Period (1975–2000)** | **Year (1975)** | **Year (2000)** | | | |
| Cropland | 559,716 | 877,396 | 317,680 | 56.8 | 2.3 |
| Forestland | 581,944 | 481,968 | −99,976 | −17.2 | −0.7 |
| Other Vegetation | 3,334,372 | 3,072,620 | −261,752 | −7.9 | −0.3 |
| Wetland | 101,176 | 102,968 | 1792 | 1.8 | 0.1 |
| Water | 46,464 | 43,880 | −2584 | −5.6 | −0.2 |
| Settlement | 14,860 | 23,952 | 9092 | 61.2 | 2.4 |
| Other LULC | 248,136 | 283,884 | 35,748 | 14.4 | 0.6 |
| Total | 4,886,668 | 4,886,668 | | | |
| **Period (2000–2013)** | **Year (2000)** | **Year (2013)** | | | |
| Cropland | 877,396 | 1,163,356 | 285,960 | 51.1 | 2.5 |
| Forestland | 481,968 | 425,260 | −56,708 | −9.7 | −0.9 |
| Other Vegetation | 3,072,620 | 2,814,152 | −258,468 | −7.8 | −0.6 |
| Wetland | 102,968 | 109,900 | 6932 | 6.9 | 0.5 |
| Water | 43,880 | 39,160 | −4720 | −10.2 | −0.8 |
| Settlement | 23,952 | 35,664 | 11,712 | 78.8 | 3.8 |
| Other LULC | 283,884 | 299,176 | 15,292 | 6.2 | 0.4 |
| Total | 4,886,668 | 4,886,668 | 0 | | |
| **Period (1975–2013)** | **Year (1975)** | **Year (2013)** | | | |
| Cropland | 559,716 | 1,163,356 | 603,640 | 107.8 | 2.8 |
| Forestland | 581,944 | 425,260 | −156,684 | −26.9 | −0.7 |
| Other Vegetation | 3,334,372 | 2,814,152 | −520,220 | −15.6 | −0.4 |
| Wetland | 101,176 | 109,900 | 8724 | 8.6 | 0.2 |
| Water | 46,464 | 39,160 | −7304 | −15.7 | −0.4 |
| Settlement | 14,860 | 35,664 | 20,804 | 140.0 | 3.7 |
| Other LULC | 248,136 | 299,176 | 51,040 | 20.6 | 0.5 |
| Total | 4,886,668 | 4,886,668 | | | |

**Table A2.** The net change in area/extent (km²) and the net relative loss/gain (%) of the seven aggregated Land Use Land Cover (LULC) classes/types in the humid and arid regions of West Africa during the 1975–2000, 2000–2013, and 1975–2013 transition periods.

| LULC Types | Area (km²) | Area (km²) | Net Loss/Gain (km²) | Net Loss/Gain (%) | Annual Rate of Change (%) |
|---|---|---|---|---|---|
| **Period (1975–2000)** | **Year (1975)** | **Year (2000)** | | | |
| **Eco-Zone** | | **Humid (Sudanian, Guinean, Guineo-Congolian)** | | | |
| Cropland | 427,736.0 | 700,084.0 | 272,348.0 | 63.7 | 2.5 |
| Forestland | 569,428.0 | 470,264.0 | −99,164.0 | −17.4 | −0.7 |
| Other Vegetation | 1,871,544.0 | 1,684,512.0 | −187,032.0 | −10.0 | −0.4 |
| Wetland | 59,348.0 | 61,524.0 | 2176.0 | 3.7 | 0.1 |
| Water | 28,900.0 | 31,696.0 | 2796.0 | 9.7 | 0.4 |
| Settlement | 13,108.0 | 21,580.0 | 8472.0 | 64.6 | 2.6 |
| Other LULC | 8244.0 | 8648.0 | 404.0 | 4.9 | 0.2 |
| Total | 2,978,308.0 | 2,978,308.0 | | | |
| **Eco-Zone** | | **Arid (Sahel)** | | | |
| Cropland | 131,596.0 | 176,644.0 | 45,048.0 | 34.2 | 1.4 |
| Forestland | 10,712.0 | 10,132.0 | −580.0 | −5.4 | −0.2 |
| Other Vegetation | 1453,828.0 | 1,379,768.0 | −74,060.0 | −5.1 | −0.2 |
| Wetland | 41,416.0 | 41,012.0 | −404.0 | −1.0 | 0.0 |
| Water | 17,344.0 | 11,956.0 | −5388.0 | −31.1 | −1.2 |
| Settlement | 1668.0 | 2212.0 | 544.0 | 32.6 | 1.3 |
| Other LULC | 193,360.0 | 228,200.0 | 3,4840.0 | 18.0 | 0.7 |
| Total | 1,849,924.0 | 1,849,924.0 | | | |
| **Period (2000–2013)** | **Year (2000)** | **Year (2013)** | | | |
| **Eco-Zone** | | **Humid (Sudanian, Guinean, Guineo-Congolian)** | | | |
| Cropland | 700,084.0 | 932,564.0 | 232,480.0 | 54.4 | 2.6 |
| Forestland | 470,264.0 | 414,464.0 | −55,800.0 | −9.8 | −0.9 |
| Other Vegetation | 1684,512.0 | 1,495,684.0 | −188,828.0 | −10.1 | −0.9 |
| Wetland | 61,524.0 | 62,308.0 | 784.0 | 1.3 | 0.1 |
| Water | 31,696.0 | 31,604.0 | −92.0 | −0.3 | 0.0 |
| Settlement | 21,580.0 | 32,148.0 | 10,568.0 | 80.6 | 3.8 |
| Other LULC | 8648.0 | 9536.0 | 888.0 | 10.8 | 0.8 |
| Total | 2,978,308.0 | 2,978,308.0 | | | |

**Table A2.** *Cont.*

| LULC Types | Area (km$^2$) | Area (km$^2$) | Net Loss/Gain (km$^2$) | Net Loss/Gain (%) | Annual Rate of Change (%) |
|---|---|---|---|---|---|
| **Eco-Zone** | | **Arid (Sahel)** | | | |
| Cropland | 176,644.0 | 229,924.0 | 53,280.0 | 40.5 | 2.3 |
| Forestland | 10,132.0 | 9316.0 | −816.0 | −7.6 | −0.6 |
| Other Vegetation | 1,379,768.0 | 1,310,980.0 | −68,788.0 | −4.7 | −0.4 |
| Wetland | 41,012.0 | 47,152.0 | 6140.0 | 14.8 | 1.2 |
| Water | 11,956.0 | 7352.0 | −4604.0 | −26.5 | −3.0 |
| Settlement | 2212.0 | 3280.0 | 1068.0 | 64.0 | 3.7 |
| Other LULC | 228,200.0 | 241,920.0 | 13,720.0 | 7.1 | 0.5 |
| Total | 1,849,924.0 | 1,849,924.0 | | | |
| **Period (1975–2013)** | **Year (1975)** | **Year (2013)** | | | |
| **Eco-Zone** | | **Humid (Sudanian, Guinean, Guineo-Congolian)** | | | |
| Cropland | 427,736.0 | 932,564.0 | 504,828.0 | 118.0 | 3.1 |
| Forestland | 569,428.0 | 414,464.0 | −154,964.0 | −27.2 | −0.7 |
| Other Vegetation | 1,871,544.0 | 1,495,684.0 | −375,860.0 | −20.1 | −0.5 |
| Wetland | 59,348.0 | 62,308.0 | 2960.0 | 5.0 | 0.1 |
| Water | 28,900.0 | 31,604.0 | 2704.0 | 9.4 | 0.2 |
| Settlement | 13,108.0 | 32,148.0 | 19,040.0 | 145.3 | 3.8 |
| Other LULC | 8244.0 | 9536.0 | 1292.0 | 15.7 | 0.4 |
| Total | 2,978,308.0 | 2978,308.0 | | | |
| **Eco-Zone** | | **Arid (Sahel)** | | | |
| Cropland | 131,596.0 | 229,924.0 | 98,328.0 | 74.7 | 2.0 |
| Forestland | 10,712.0 | 9316.0 | −1396.0 | −13.0 | −0.3 |
| Other Vegetation | 1,453,828.0 | 1,310,980.0 | −142,848.0 | −9.8 | −0.3 |
| Wetland | 41,416.0 | 47,152.0 | 5736.0 | 13.8 | 0.4 |
| Water | 17,344.0 | 7352.0 | −9992.0 | −57.6 | −1.5 |
| Settlement | 1668.0 | 3280.0 | 1612.0 | 96.6 | 2.5 |
| Other LULC | 193,360.0 | 241,920.0 | 48,560.0 | 25.1 | 0.7 |
| Total | 1,849,924.0 | 18,49,924.0 | | | |

**Table A3.** The Land Use Land Cover (LULC) transitions matrix for the humid and the arid regions of West Africa in (1975–2000, 2000–2013, and 1975–2013). Changes by class are calculated by dividing the area in each off-diagonal element by the area of the class indicated in the left-most column (i.e., initial reference year of each period) in each block corresponding to each region in Table A2.

| LULC Types | Cropland Area (km²) | % | Forestland Area (km²) | % | Other Vegetation Area (km²) | % | Wetland Area (km²) | % | Water Area (km²) | % | Settlement Area (km²) | % | Other LULC Area (km²) | % |
|---|---|---|---|---|---|---|---|---|---|---|---|---|---|---|
| **Period (1975–2000)** | | | | | | | | | | | | | | |
| **Eco-Zone** | | | | | **Humid (Sudanian, Guinean, Guineo-Congolian)** | | | | | | | | | |
| Cropland | 215,956.0 | 50.5 | 41,084.0 | 9.6 | 151,876.0 | 35.5 | 8000.0 | 1.9 | 2984.0 | 0.7 | 6652.0 | 1.6 | 1184.0 | 0.3 |
| Forestland | 90,344.0 | 15.9 | 289,144.0 | 50.8 | 175,112.0 | 30.8 | 3772.0 | 0.7 | 5016.0 | 0.9 | 5116.0 | 0.9 | 924.0 | 0.2 |
| Other Vegetation | 372,788.0 | 19.9 | 129,836.0 | 6.9 | 1,322,168.0 | 70.6 | 27,332.0 | 1.5 | 8776.0 | 0.5 | 6060.0 | 0.3 | 4584.0 | 0.2 |
| Wetland | 11,996.0 | 20.2 | 2640.0 | 4.4 | 22,456.0 | 37.8 | 19,020.0 | 32.0 | 2740.0 | 4.6 | 260.0 | 0.4 | 236.0 | 0.4 |
| Water | 3040.0 | 10.5 | 4376.0 | 15.1 | 6300.0 | 21.8 | 2932.0 | 10.1 | 11,812.0 | 40.9 | 324.0 | 1.1 | 116.0 | 0.4 |
| Settlement | 4436.0 | 33.8 | 2508.0 | 19.1 | 2520.0 | 19.2 | 236.0 | 1.8 | 232.0 | 1.8 | 3120.0 | 23.8 | 56.0 | 0.4 |
| Other LULC | 1524.0 | 18.5 | 676.0 | 8.2 | 4080.0 | 49.5 | 232.0 | 2.8 | 136.0 | 1.6 | 48.0 | 0.6 | 1548.0 | 18.8 |
| Total | 700,084.0 | | 470,264.0 | | 1,684,512.0 | | 61,524.0 | | 31,696.0 | | 21,580.0 | | 8648.0 | |
| **Eco-Zone** | | | | | **Arid (Sahel)** | | | | | | | | | |
| Cropland | 85,412.0 | 64.9 | 356.0 | 0.3 | 39,048.0 | 29.7 | 2280.0 | 1.7 | 772.0 | 0.6 | 1392.0 | 1.1 | 2336.0 | 1.8 |
| Forestland | 576.0 | 5.4 | 3188.0 | 29.8 | 5440.0 | 50.8 | 420.0 | 3.9 | 96.0 | 0.9 | 24.0 | 0.2 | 968.0 | 9.0 |
| Other Vegetation | 82,336.0 | 5.7 | 5388.0 | 0.4 | 1,247,956.0 | 85.8 | 15,216.0 | 1.0 | 3352.0 | 0.2 | 492.0 | 0.0 | 99,088.0 | 6.8 |
| Wetland | 3212.0 | 7.8 | 320.0 | 0.8 | 16,848.0 | 40.7 | 15,616.0 | 37.7 | 2892.0 | 7.0 | 60.0 | 0.1 | 2468.0 | 6.0 |
| Water | 584.0 | 3.4 | 64.0 | 0.4 | 6424.0 | 37.0 | 5296.0 | 30.5 | 4500.0 | 25.9 | 24.0 | 0.1 | 452.0 | 2.6 |
| Settlement | 1136.0 | 68.1 | 4.0 | 0.2 | 240.0 | 14.4 | 64.0 | 3.8 | 24.0 | 1.4 | 172.0 | 10.3 | 28.0 | 1.7 |
| Other LULC | 3388.0 | 1.8 | 812.0 | 0.4 | 63,812.0 | 33.0 | 2120.0 | 1.1 | 320.0 | 0.2 | 48.0 | 0.0 | 122,860.0 | 63.5 |
| Total | 176,644.0 | | 10,132.0 | | 1,379,768.0 | | 41,012.0 | | 11,956.0 | | 2212.0 | | 228,200.0 | |
| **Period (2000–2013)** | | | | | | | | | | | | | | |
| **Eco-Zone** | | | | | **Humid (Sudanian, Guinean, Guineo-Congolian)** | | | | | | | | | |
| Cropland | 641,984.0 | 91.7 | 6596.0 | 0.9 | 45,160.0 | 6.5 | 1116.0 | 0.2 | 172.0 | 0.0 | 4828.0 | 0.7 | 228.0 | 0.0 |
| Forestland | 33,472.0 | 7.1 | 401,280.0 | 85.3 | 31,848.0 | 6.8 | 1104.0 | 0.2 | 300.0 | 0.1 | 2052.0 | 0.4 | 208.0 | 0.0 |
| Other Vegetation | 251,964.0 | 15.0 | 6132.0 | 0.4 | 1,415,528.0 | 84.0 | 5000.0 | 0.3 | 1252.0 | 0.1 | 3788.0 | 0.2 | 848.0 | 0.1 |
| Wetland | 4096.0 | 6.7 | 336.0 | 0.5 | 2396.0 | 3.9 | 53,388.0 | 86.8 | 1140.0 | 1.9 | 120.0 | 0.2 | 48.0 | 0.1 |
| Water | 800.0 | 2.5 | 56.0 | 0.2 | 468.0 | 1.5 | 1620.0 | 5.1 | 28,668.0 | 90.4 | 8.0 | 0.0 | 76.0 | 0.2 |
| Settlement | 156.0 | 0.7 | 52.0 | 0.2 | 48.0 | 0.2 | 0.0 | 0.0 | 8.0 | 0.0 | 21,308.0 | 98.7 | 8.0 | 0.0 |
| Other LULC | 92.0 | 1.1 | 12.0 | 0.1 | 236.0 | 2.7 | 80.0 | 0.9 | 64.0 | 0.7 | 44.0 | 0.5 | 8120.0 | 93.9 |
| Total | 932,564.0 | | 414,464.0 | | 1,495,684.0 | | 62,308.0 | | 31,604.0 | | 32,148.0 | | 9536.0 | |

**Table A3.** *Cont.*

| LULC Types | Cropland | | Forestland | | Other Vegetation | | Wetland | | Water | | Settlement | | Other LULC | |
|---|---|---|---|---|---|---|---|---|---|---|---|---|---|---|
| | Area (km$^2$) | % | Area (km$^2$) | % | Area (km$^2$) | % | Area (km$^2$) | % | Area (km$^2$) | % | Area (km$^2$) | % | Area (km$^2$) | % |
| **Eco-Zone** | | | | | **Arid (Sahel)** | | | | | | | | | |
| Cropland | 16,3272.0 | 92.4 | 0.0 | 0.0 | 11,896.0 | 6.7 | 692.0 | 0.4 | 32.0 | 0.0 | 644.0 | 0.4 | 108.0 | 0.1 |
| Forestland | 204.0 | 2.0 | 9280.0 | 91.6 | 400.0 | 3.9 | 156.0 | 1.5 | 88.0 | 0.9 | 4.0 | 0.0 | 0.0 | 0.0 |
| Other Vegetation | 62,336.0 | 4.5 | 24.0 | 0.0 | 1,294,112.0 | 93.8 | 6444.0 | 0.5 | 696.0 | 0.1 | 612.0 | 0.0 | 15,544.0 | 1.1 |
| Wetland | 2808.0 | 6.8 | 12.0 | 0.0 | 2132.0 | 5.2 | 35,336.0 | 86.2 | 644.0 | 1.6 | 36.0 | 0.1 | 44.0 | 0.1 |
| Water | 644.0 | 5.4 | 0.0 | 0.0 | 1052.0 | 8.8 | 4368.0 | 36.5 | 5876.0 | 49.1 | 12.0 | 0.1 | 4.0 | 0.0 |
| Settlement | 252.0 | 11.4 | 0.0 | 0.0 | 16.0 | 0.7 | 0.0 | 0.0 | 0.0 | 0.0 | 1940.0 | 87.7 | 4.0 | 0.2 |
| Other LULC | 408.0 | 0.2 | 0.0 | 0.0 | 1372.0 | 0.6 | 156.0 | 0.1 | 16.0 | 0.0 | 32.0 | 0.0 | 226,216.0 | 99.1 |
| Total | 229,924.0 | | 9316.0 | | 1,310,980.0 | | 47,152.0 | | 7352.0 | | 3280.0 | | 241,920.0 | |
| **Period (1975–2013)** | | | | | | | | | | | | | | |
| **Eco-Zone** | | | | | **Humid (Sudanian, Guinean, Guineo-Congolian)** | | | | | | | | | |
| Cropland | 244,472.0 | 57.2 | 36,340.0 | 8.5 | 125,076.0 | 29.2 | 7776.0 | 1.8 | 2964.0 | 0.7 | 9740.0 | 2.3 | 1368.0 | 0.3 |
| Forestland | 124,740.0 | 21.9 | 255,224.0 | 44.8 | 171,432.0 | 30.1 | 3712.0 | 0.7 | 5236.0 | 0.9 | 8004.0 | 1.4 | 1080.0 | 0.2 |
| Other Vegetation | 537,080.0 | 28.7 | 113,792.0 | 6.1 | 1,168,412.0 | 62.4 | 28,832.0 | 1.5 | 8644.0 | 0.5 | 9672.0 | 0.5 | 5112.0 | 0.3 |
| Wetland | 15,720.0 | 26.5 | 2380.0 | 4.0 | 19,688.0 | 33.2 | 18,376.0 | 31.0 | 2508.0 | 4.2 | 424.0 | 0.7 | 252.0 | 0.4 |
| Water | 3944.0 | 13.6 | 4036.0 | 14.0 | 5352.0 | 18.5 | 3172.0 | 11.0 | 11,848.0 | 41.0 | 420.0 | 1.5 | 128.0 | 0.4 |
| Settlement | 4536.0 | 34.6 | 2104.0 | 16.1 | 2136.0 | 16.3 | 212.0 | 1.6 | 244.0 | 1.9 | 3804.0 | 29.0 | 72.0 | 0.5 |
| Other LULC | 2072.0 | 25.1 | 588.0 | 7.1 | 3588.0 | 43.5 | 228.0 | 2.8 | 160.0 | 1.9 | 84.0 | 1.0 | 1524.0 | 18.5 |
| Total | 932,564.0 | | 414,464.0 | | 1,495,684.0 | | 62,308.0 | | 31,604.0 | | 32,148.0 | | 9536.0 | |
| **Eco-Zone** | | | | | **Arid (Sahel)** | | | | | | | | | |
| Cropland | 94,692.0 | 72.0 | 328.0 | 0.2 | 29,392.0 | 22.3 | 2668.0 | 2.0 | 532.0 | 0.4 | 1772.0 | 1.3 | 2212.0 | 1.7 |
| Forestland | 792.0 | 7.4 | 3064.0 | 28.6 | 5260.0 | 49.1 | 492.0 | 4.6 | 88.0 | 0.8 | 36.0 | 0.3 | 980.0 | 9.1 |
| Other Vegetation | 121,484.0 | 8.4 | 4816.0 | 0.3 | 1,193,264.0 | 82.1 | 18,788.0 | 1.3 | 2436.0 | 0.2 | 996.0 | 0.1 | 112,044.0 | 7.7 |
| Wetland | 5732.0 | 13.8 | 292.0 | 0.7 | 15,204.0 | 36.7 | 16,152.0 | 39.0 | 1388.0 | 3.4 | 112.0 | 0.3 | 2536.0 | 6.1 |
| Water | 1136.0 | 6.5 | 44.0 | 0.3 | 6440.0 | 37.1 | 6588.0 | 38.0 | 2656.0 | 15.3 | 56.0 | 0.3 | 424.0 | 2.4 |
| Settlement | 1168.0 | 70.0 | 4.0 | 0.2 | 164.0 | 9.8 | 60.0 | 3.6 | 16.0 | 1.0 | 224.0 | 13.4 | 32.0 | 1.9 |
| Other LULC | 4920.0 | 2.5 | 768.0 | 0.4 | 61,256.0 | 31.7 | 2404.0 | 1.2 | 236.0 | 0.1 | 84.0 | 0.0 | 123,692.0 | 64.0 |
| Total | 229,924.0 | | 9316.0 | | 1,310,980.0 | | 47,152.0 | | 7352.0 | | 3280.0 | | 241,920.0 | |

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
