# Peer review of "Mapping Land Use Land Cover Transitions at Different Spatiotemporal Scales in West Africa"

_sustainability, doi:10.3390/su12208565_

Round 1

Reviewer 1 Report

The paper is undoubtedly topical. The authors used more than 100 references and sources. They analysed the land use / land cover changes in different regional structures and documented them graphically. The paper is methodologically very well based. Its conclusion could inspire further investigation in the field.

On the other side, the authors consider the increase of settlements for the most important land use change (also for the next research). Although the area of human settlements has arisen more than twice during the period under investigation, the settlements in the area under study take less than 1% - which is very few to compare with Europe. It means that the widening of Africa´s cities can hardly play any important role in global environmental change of West Africa at the moment. Moreover, the pressure of urbanization on the natural vegetation will continue also in the future. Women of West Africa have the highest fertility rates in the World, which reaches round 5 children per women. Even if values of this indicator will certainly decrease, the young population structure of this part of Africa will still reproduce during some next generations. The area of settlements will increase not only because of the population growth only, but also due to the presupposed increasing demand on the living area per inhabitant, demond on infrastructure etc., connected with the growth of the middle class. Whether the process is environmentally efficient, it is another question similarly as uncontrolled growth of the biggest cities. However, it is the question of debates on local (national) levels.

The main driver is evidently the change of natural vegetation into artificial land use. This is the process which Europe passed centuries ago. However, it is not necessary to repeat Europe´s mistakes. Nevertheless, if we want to discuss a wider context, we should mention that countries of West Africa have limited possibilities how to participate in the World market. That is why they have to export their natural sources – minerals, timber, crops growing in the plantation way. Thus in fact they export their natural resources, which surely has serious environmental impacts. To solve this hardly solvable problem is the only way how to stop the degradation of the vegetation in West Africa globally. However, the article can be used as a warning prognosis.

Particular solutions consist in local measures – how efficiently are natural sources of West Africa used to gain the highest benefit under lowest environmental damage.

The paper is too long and verbose. Precision of explanations comes at the expense of the readability. Some information repeats more times. The investigation finished by the year 2013 which means that the last results are 7 years old. Is there some newer information at disposal?

In the line 488, 1975 should be the right year.

Author Response

Response to Comments by Reviewer 1

Point 1: The paper is undoubtedly topical. The authors used more than 100 references and sources. They analyzed the land use / land cover changes in different regional structures and documented them graphically. The paper is methodologically very well based. Its conclusion could inspire further investigation in the field.

Response 1: Thank you very much for the review, comments and suggestions. We appreciate your time and inputs.

Point 2: On the other side, the authors consider the increase of settlements for the most important land use change (also for the next research). Although the area of human settlements has arisen more than twice during the period under investigation, the settlements in the area under study take less than 1% - which is very few to compare with Europe. It means that the widening of Africa´s cities can hardly play any important role in global environmental change of West Africa at the moment. Moreover, the pressure of urbanization on the natural vegetation will continue also in the future. Women of West Africa have the highest fertility rates in the World, which reaches round 5 children per women. Even if values of this indicator will certainly decrease, the young population structure of this part of Africa will still reproduce during some next generations. The area of settlements will increase not only because of the population growth only, but also due to the presupposed increasing demand on the living area per inhabitant, demand on infrastructure etc., connected with the growth of the middle class. Whether the process is environmentally efficient, it is another question similarly as uncontrolled growth of the biggest cities. However, it is the question of debates on local (national) levels.

Response 2: Many thanks for this comment. We aimed to emphasise on the concurrent expansion of human-managed Land Use Land Cover (LULC, e.g., cropland and settlements).  In our view, this is an indication of the quest to expand crop production and accommodation to support the growing population in West Africa. On the hand, we agree with your comments and suggestions. In addition to the reasons you cited, another possible reason for the relatively low fractional abundance of settlement in West Africa could be the extended family system in Africa which permits a considerable number of the family members to live in the same compound house. This kind of family system increases the total number of inhabitant per unit area and may lead to urbanisation and subsequent urban sprawl in the Sub-continent. This process in turn has the tendency to put pressure on the natural resources (Flückiger and Ludwig, 2017; Güneralp et al., 2017). The manuscript has been revised to incorporate your suggestions in our discussion.

Changes:

L1047-L1063, Sect.5.1; revised manuscript: we have explained the implication of the past and present settlement expansions in West Africa and the possible reasons why the fractional abundance is significantly low at the moment as well as other factors which may influence future expansions.

Point 3: The main driver is evidently the change of natural vegetation into artificial land use. This is the process which Europe passed centuries ago. However, it is not necessary to repeat Europe´s mistakes. Nevertheless, if we want to discuss a wider context, we should mention that countries of West Africa have limited possibilities how to participate in the World market. That is why they have to export their natural sources–minerals, timber, crops growing in the plantation way. Thus, in fact they export their natural resources, which surely have serious environmental impacts. To solve this hardly solvable problem is the only way how to stop the degradation of the vegetation in West Africa globally. However, the article can be used as a warning prognosis.

Response 3: Many thanks for this comment. We believe that an additional study would be needed, however, to articulate a policy option based on our findings. We have endeavoured to suggest how our findings could be applied in that direction.

Changes:

L1032-L1036, L1042-1046, Sect. 5.1; revised manuscript:  We have highlighted the implications of our findings and the policy options to mitigate the degradation of natural vegetation in West Africa.

Point 4: Particular solutions consist in local measures – how efficiently are natural sources of West Africa used to gain the highest benefit under lowest environmental damage.

Response 4: Many thanks for this comment.  

Our findings may be used to identify priorities to mitigate natural resources degradation through LULC change management in West Africa. One of the ways to tackle the degradation of natural resources in Africa is to strengthen the agriculture value chain and adoption of natural resource regeneration technologies. According to Gnansounou et al. (2020), natural resource regeneration technologies such as bio-energy technologies will ease the pressure on hydro-power generation from natural water bodies to ensure maximum benefit under minimal environmental damage.

Changes:

L1042-1046, Sect. 5.1; revised manuscript: The above points have been highlighted.

L1159-1172, Sect.5.2; revised manuscript: We cited the successful implementation of “Farmer-Led Natural Re-generation of Trees” which is a natural resource resilience building initiative implemented in Maradi and Zinder regions of Niger, Central Plateau in Burkina Faso and Seno Plain in Mali. According to Tappan and Mcgahuey (2007) ; Reij et al. (2009) and Sendzimir et al. (2011), these interventions transformed degraded landscapes into fertile land for farming, which in turn increased drought resistance, enhanced crop productivity, improved food security and alleviated poverty in the mid-1980s.

Point 5: The paper is too long and verbose. Precision of explanations comes at the expense of the readability. Some information repeats more times.

Response 5: Many thanks for this comment. We have streamlined the revised manuscript.

Changes:

 L1385-L 1466, Sect.6; revised manuscript

  • The summary part of the “Summary and Conclusion” section has been deleted
  • The conclusion has been narrowed and shortened significantly.
  • Redundant text has been deleted wherever possible.

Point 6: The investigation finished by the year 2013 which means that the last results are 7 years old. Is there some newer information at disposal?

Response 6:  Thanks for this comment.

Recently, many  Earth Observation (EO) LULC data which extend beyond the temporal coverage of our current study are available at the continental and global scales. However, satellite imagery from different sources and different approaches were used to generate these LULC data. According to Tsendbazar et al. (2016), LULC statistics from multiple sources are inconsistent at the moment. One of our main foci was to examine the Land Use Land Cover (LULC) transitions during the drought period in West Africa. The LULC data by CILSS (2016) are the only data which cover the temporal resolution of the drought period (i.e., 70s) in West Africa.

Moreover, forestland and cropland estimates from the USGS LULC map in the year 2000 had been evaluated against FAOSTATS and other global LULC maps by our research team. We found that the forestland and cropland estimates of the USGS LULC data agreed better with the FAOSTATS as compared to the other global LULC data. Therefore, our analyses were only based on the period of time covered by CILSS (2016) to avoid uncertainty surrounding the data gaps and the LULC estimates from different sources.

Changes

The above points have been highlighted in the manuscript (see L 217-225 and L227-230).

Point 7: In the line 488, 1975 should be the right year.

Response 7: Thanks for this comment. The correction has been done as suggested (see L500).

Reviewer 2 Report

The paper is well-structured and the results are interesting. Nevertheless, before its publication, some minor issues should be solved.

The Abstract is too long and should be shortened.

Some references should be included in the introduction section for being important in the African Development Agenda, such as

https://doi.org/10.3390/ijgi8070292
https://doi.org/10.3390/ijgi8090399

Regarding English language use:

-Please, you should write the paper in one of them (American or British English). In the text, it appears "analyzing" and "analysing"; "categorized" and "categorised"; "fertilizers" and "fertilisers"; "urbanization" and "urbanisation", and so forth.

Before "i.e." or "e.g.", a comma "," must be written.

L42: A comma "," is missing in "Africa, especially the Shale region is often..." and similar sentences. Please, rewrite as "Africa, especially the Sahel region, is often...".

There are some grammatical errors related to the use of articles "the" and "a". Sometimes an article is missing ("triggered environmental change", " with relative net gain", and so forth), and other times must be removed ("the entire West Africa").

The hyphen is missing in most of cases: "built up", "sample based", "open source", "human induced", "post drought", "sub regions", and so forth.

To define year interval a hyphen "1975-2013" is used instead of an en dash "1975—2013".

Please, use "and so forth" instead of "etc".

Please, write "seven" or "six" rather than "7" or "6", and so forth.

L95: Geo-Information Science (GIS) and L125: "Geo-Information System (GIS)". Later on, "GIS" acronym is used, and it is not clear whether "S" refers to Science or System. Please, do not write the acronym in one of them, L95 or L125, depending on the rest of the text.

L251: "pre-cambrian" should be replaced with "Pre-Cambrian".

L359: "[76,77-78]" should be replaced with "[76-78]".

L477: "Y 2" should be replaced with "Y2".

Author Response

Response to Comments by Reviewer 2

Point 1: The paper is well-structured and the results are interesting. Nevertheless, before its publication, some minor issues should be solved.

Response 1: Thank you very much for the review, comments and suggestions. We appreciate your time and inputs.

Point 2: The Abstract is too long and should be shortened.

Response 2:  Thanks for this comment. The Abstract has been shortened wherever possible and streamlined (see L15-21, 32-33).

Point 3: Some references should be included in the introduction section for being important in the African Development Agenda, such as

https://doi.org/10.3390/ijgi8070292
https://doi.org/10.3390/ijgi8090399

Response 3: Thanks for this comment. Modified as suggested. References have been inserted at L166-L173.

Point 4: Regarding English language use: - Please, you should write the paper in one of them (American or British English). In the text, it appears "analyzing" and "analysing"; "categorized" and "categorised"; "fertilizers" and "fertilisers"; "urbanization" and "urbanisation", and so forth. Before "i.e." or "e.g.", a comma "," must be written.

Response 4: Thanks for this comment. Modified as suggested.

 Changes:

Spelling has been modified to use British English, see L135, L214, L265, L279, L281, L265, L432, L452, L458, L462, L465, L504, L524, L667, L1193 and so forth.

A comma has been inserted at L42, L59, L64, L89, L111, L290, L328, L345, L371, L401 L404, L433, L449, L463, L468, L476, L532, L641, L649, L651, L653, L692, L802, L829, L864-L865, L867, L870, L890, L899, L903, L906, L909, L1067, L1072, L1120, L1311 and so forth.

Point 5: L42: A comma "," is missing in "Africa, especially the Sahel region is often..." and similar sentences. Please, rewrite as "Africa, especially the Sahel region, is often...”

Response 5:  Thanks for this comment.  Modified as suggested (see L42).

Point 6: There are some grammatical errors related to the use of articles "the" and "a". Sometimes an article is missing ("triggered environmental change", “with relative net gain", and so forth), and other times must be removed ("the entire West Africa").

Response 6:  Thanks for this comment. Modified as suggested.

Changes:

Articles have been corrected at L601, L602, L603, L620, L644, L662, L701 L706, L714, L752, L764, L772, L778, L791, L805, L827, L843, L848, L856, L903, L1146, L1274 and so forth.

Point 7 : The hyphen is missing in most of cases: "built up", "sample based", "open source", "human induced", "post drought", "sub regions", and so forth.

Response 7: Thanks for this comment. Modified as suggested.

Changes:

Hyphen added at L121, L136, L157, L160, L251, L209, L L210, L216, L342, L349, L356 L386, L387, L467, L469, L540, L764, L831, L860, L 865, and L1475 and so on.

Point 8: To define year interval a hyphen "1975-2013" is used instead of an en dash "1975—2013".

Response 8:  Thanks for this comment. Modified as suggested.

Point 9: Please, use "and so forth" instead of "etc".

Response 9:  Thanks for this comment. Modified as suggested.

Changes:

“and so forth” inserted at L130, L319, L394,  L642, L1148 and so on.

Point 10: Please, write "seven" or "six" rather than "7" or "6", and so forth.

Response 10:  Thanks for this comment. Modified as suggested except when dates are being indicated.

Changes:

Numbers replaced at L231, L312, L313, L323, L334, L338, L343, L 345, L354, L367, L399, L402, L410, L432, L443, L445, L449, L454, L477, L499, L500, L562, L564, L566, L567, L595, L618, L619, L668, L781 and so forth.

Point 11: L95: Geo-Information Science (GIS) and L125: "Geo-Information System (GIS)". Later on, "GIS" acronym is used, and it is not clear whether "S" refers to Science or System. Please, do not write the acronym in one of them, L95 or L125, depending on the rest of the text.

Response 11: “Geo-Information Science (GIS)” at L95 of the original manuscript has been changed to “Geo-Information Science (GISc) in the revised manuscript whilst “Geo-Information System (GIS)” at L125 has been maintained.

Point 12: L251: "pre-cambrian" should be replaced with "Pre-Cambrian".

Response 12:  Thanks for this comment. Modified as suggested (see L268).

Point 13:L359: "[76, 77-78]" should be replaced with "[76-78]".

Response 13: Thanks for this comment. Modified as suggested (now, “[81-83]” at L376) after the revision.

Point 14: L477: "Y 2" should be replaced with "Y2".

Response 14:  Thanks for this comment. Modified as suggested (now, see L489).

Round 2

Reviewer 1 Report

In my opinion, the explanations are satisfactory and the changes made in the manuscript are sufficient